# Spectral Heat Flow for Conservative Token Condensation in Vision-Language Models

Zhaoyang Li [1] [*]  Yanjun Li [1] [*]  Wangkai Li [1]  Yujia Chen [1]  Tianzhu Zhang [1]

## Abstract

Vision-Language Models (VLMs) are costly at inference time because they must process long sequences of visual tokens. Existing token pruning methods often degrade under high compression by blindly discarding information, breaking spatial structure or collapsing diversity. We propose SpecFlow, a training-free framework that shifts the paradigm from destructive pruning to conservative condensation, strictly enforcing spatial coverage and statistical conservation to ensure stability. Treating visual tokens as nodes in a $k$NN graph, SpecFlow (i) computes a stable importance field via spectral heat flow to preserve structural coherence, (ii) allocates budgets via adaptive spatial partitioning to guarantee coverage, and (iii) aggregates discarded information into coreset sinks to maintain statistical conservation. The method is plug-and-play, requires no fine-tuning, and is compatible with FlashAttention. Experiments confirm that our SpecFlow outperforms SOTA methods across tasks, VLM architectures, and pruning ratios. Notably, LLaVA-1.5 with SpecFlow retains 95.6% of original performance despite pruning 88.9% of visual tokens, offering an exceptional efficiency-accuracy balance. Code is available at https://github.com/Lzy-dot/SpecFlow.

## 1. Introduction

Recent Vision-language models (VLMs) (Li et al., 2025a; Team et al., 2023; Liu et al., 2024c; Chen et al., 2024c) have shown remarkable progress in multimodal understanding, delivering strong results on visual question answering (Guo

[*]Equal contribution [1]School of Information Science and Technology / National Key Laboratory of Deep Space Exploration, University of Science and Technology of China. Correspondence to: Tianzhu Zhang <tzzhang@ustc.edu.cn>.

*Proceedings of the $43^{rd}$ International Conference on Machine Learning*, Seoul, South Korea. PMLR 306, 2026. Copyright 2026 by the author(s).

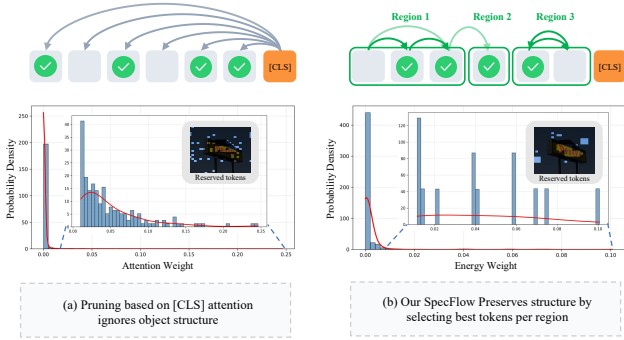

*Figure 1.* **Global Top-$K$ pruning vs. our proposed SpecFlow.** (a) Global Top-$K$ pruning based on `[CLS]` attention can yield fragmented selections due to spiky attention distributions, often creating spatial holes within objects. (b) SpecFlow diffuses attention-derived energy on a $k$NN token graph and applies coverage-aware regional budgeting, yielding region-coherent retained tokens under high compression.

et al., 2023; Zhao et al., 2024; Huynh et al., 2025; Kuang et al., 2025), image captioning (Ghandi et al., 2023; Chen et al., 2024a), and video understanding (Lin et al., 2024; Maaz et al., 2024; Wang et al., 2024b).Despite this progress, efficient deployment remains challenging due to the large number of visual tokens produced by modern vision encoders. For example, LLaVA-1.5 (Liu et al., 2024a) typically encodes a $336 \times 336$ image into 576 patch tokens, which are then processed by the language model during the prefill stage, often resulting in substantially higher latency than text-only inference.

A natural approach to accelerate inference is to reduce the number of visual tokens. Existing token pruning methods (Chen et al., 2024b; Yang et al., 2025; Zhang et al., 2024b; Zou et al., 2025) commonly perform token pruning via Top-$K$ selection using attention-derived scores. While effective in reducing computation, such point-wise ranking can be brittle under high compression because it disregards two properties of visual token sets: spatial coherence (tokens corresponding to a region tend to form contiguous structures) and contextual dependency (background and surrounding regions can be essential for reasoning). Empirically, these limitations manifest as two recurring failure modes. First, attention scores can be highly concentrated, so Top-$K$ selection often yields spatially fragmented token subsets, creating "holes" within objects and disrupting

coherent visual evidence (Figure 1 (a)). Second, saliency-driven pruning tends to over-preserve foreground regions while aggressively removing contextual background, which can impair tasks that rely on spatial relations and scene-level semantics.

These observations suggest that token pruning should not be treated solely as independent, discrete selection. Instead, an effective pruning rule should respect the collective structure of visual tokens: importance should vary smoothly across coherent regions, while allocation should maintain coverage to avoid systematic blind spots. Motivated by this, we reformulate visual token pruning as an importance propagation problem on a token graph (Figure 1 (b)).

We propose SpecFlow, a training-free framework for efficient VLM inference that performs structure-preserving token pruning via spectral heat diffusion (Coifman & Lafon, 2006). SpecFlow constructs a graph over visual tokens and computes a stable importance field by propagating initial saliency through spectral heat flow, which naturally smooths spiky attention into region-consistent importance. To prevent spatial collapse under high compression, SpecFlow further allocates per-region budgets using adaptive quadtree partitioning, enforcing explicit spatial coverage. Finally, rather than discarding information, SpecFlow conservatively aggregates pruned tokens into coreset "sinks", preserving summary statistics and diversity among the retained representations. SpecFlow is plug-and-play, requires no fine-tuning, and is compatible with efficient attention implementations such as FlashAttention (Dao et al., 2022; Dao, 2023).

In summary, our contributions to the community include:

1. We characterize two common failure modes of attention-score-based visual token pruning for VLMs: spatial fragmentation and loss of contextual evidence, especially under high compression.

2. We introduce spectral heat flow on a $k$NN token graph as a training-free importance propagation mechanism, producing a stable and region-coherent importance field for pruning.

3. We propose coverage-aware pruning via adaptive quadtree partitioning with energy-proportional budget allocation, and a conservative condensation step that aggregates removed tokens into a single coreset sink (with an additional diversity-preserving anchor token).

## 2. Related Work

### 2.1. Vision-Language Models

Recent progress in large language models has spurred vision-language models (VLMs) that transfer strong language capabilities to multimodal understanding and generation. Many

recent VLMs (Liu et al., 2024a; Bai et al., 2023; Li et al., 2025b;c; Liu et al., 2024c; Lin et al., 2024; Liu et al., 2024b) improve accuracy by increasing the visual token budget through higher image resolution or multiple frames. However, the resulting visual sequences grow rapidly in length and introduce substantial computational overhead. For example, LLaVA-1.5 (Liu et al., 2024a) encodes a $336 \times 336$ image into 576 tokens, and LLaVA-NeXT (Liu et al., 2024b) scales to 2,880 tokens at $672 \times 672$. Video models such as VideoLLaVA (Lin et al., 2024) and VideoPoet (Kondratyuk et al., 2023) further require thousands of tokens to represent multiple frames. Moreover, simply increasing the token budget does not fully eliminate visual deficiencies and hallucinations, while further amplifying inference cost. These challenges motivate token-efficient visual representations and sparsification methods that substantially reduce computation while maintaining comparable performance.

### 2.2. Visual Token Compression for VLMs

Visual token compression is crucial for scaling VLMs to high-resolution images and long videos under limited compute and context. Prior work mainly follows two routes. The first is pre-LLM compression, which reduces visual tokens before they enter the decoder. For example, LLaMA-VID (Li et al., 2024) adopts a compact frame representation, and VisionZip (Yang et al., 2025) selects dominant tokens using vision-encoder attention and can merge the remaining tokens. The second is decoder-stage sparsification, which prunes visual tokens during LLM inference based on their diminishing utility in deeper layers. In this category, FastV (Chen et al., 2024b) prunes low-importance visual tokens after an early layer using attention-based ranking in a plug-and-play manner, while SparseVLM (Zhang et al., 2024b) further introduces training-free, text-guided pruning with adaptive per-layer sparsity. Despite their effectiveness, training-free pruning often relies on spiky attention scores and some methods are not compatible with FlashAttention (Dao et al., 2022; Dao, 2023). Our SpecFlow instead diffuses saliency to produce region-consistent importance and adopts a FlashAttention-compatible pruning design.

## 3. Preliminary and Motivation

### 3.1. Preliminary

**VLM inference and computation complexity.** We consider a VLM $\mathcal{M}_\theta$ composed of a vision encoder $f_{\text{vis}}$, a vision–language projection module $g$, and an autoregressive decoder $f_{\text{lm}}$ with $L$ Transformer layers. Given an image $I$ and a text prompt $X = (x_1, \ldots, x_T)$, the vision encoder produces patch features $V = f_{\text{vis}}(I) \in \mathbb{R}^{N \times d_v}$, which are mapped to language-aligned visual tokens $Z = g(V) \in \mathbb{R}^{N \times d}$. The decoder processes the concatenated sequence $[Z; E(X)]$ during prefill and then generates outputs autore-

gressively. The total decoder input length is $n = N + T$, where $T = n_{\text{sys}} + n_{\text{q}}$ denotes the number of text tokens including the system prompt and the user query. Since Transformer self-attention in prefill scales quadratically (Vaswani et al., 2017) with the sequence length, the dominant cost is

$$\mathcal{C}_{\text{prefill}} = \mathcal{O}\left(L \cdot n^2 \cdot d\right), \tag{1}$$

where $d$ is the hidden dimension. In typical VLM settings, $N \gg T$, so computation is largely dominated by visual tokens, making the reduction of $N$ essential for improving VLM inference efficiency.

**Graph diffusion for token importance.** Let $G = (\mathcal{V}, \mathcal{E})$ be a graph over $N$ tokens and let $W \in \mathbb{R}^{N \times N}$ be a nonnegative row-stochastic matrix. Given an initial importance signal $e^{(0)} \in \mathbb{R}^N$, a standard way to incorporate neighborhood consistency is the following diffusion update:

$$e^{(t+1)} = (1 - \alpha)e^{(0)} + \alpha W^\top e^{(t)}, \tag{2}$$

where $\alpha \in (0, 1)$ controls the propagation strength. This process is closely related to personalized PageRank and yields a smoothed importance estimate (Page et al., 1999; Jeh & Widom, 2003).

**Proposition 3.1** (Restart diffusion as Dirichlet-regularized smoothing). *Assume the transition matrix $W$ is obtained by row-normalizing a symmetric affinity matrix $\mathbf{S} \in \mathbb{R}^{N \times N}$, i.e., $W = D^{-1}\mathbf{S}$ where $\mathbf{S} = \mathbf{S}^\top \geq 0$ and $D = \text{diag}(\mathbf{S1})$. Then the diffusion in Eq. (2) converges to a unique fixed point $e^\star$ satisfying*

$$(\mathbf{I} - \alpha W^\top)e^\star = (1 - \alpha)e^{(0)}. \tag{3}$$

*Moreover, $e^\star$ is equivalently characterized as the unique minimizer of a convex objective that balances fidelity to the seed and graph smoothness:*

$$\begin{aligned} e^\star = \arg\min_{e \in \mathbb{R}^N} \ & \frac{1}{2}\left\|D^{-1}e - D^{-1}e^{(0)}\right\|_D^2 \\ & + \frac{\alpha}{2(1 - \alpha)}(D^{-1}e)^\top(D - \mathbf{S})(D^{-1}e). \end{aligned} \tag{4}$$

*where $\|u\|_D^2 = u^\top D u$ and $(D - \mathbf{S})$ is the (combinatorial) graph Laplacian.*

Proof and additional discussion are deferred to Appendix A.1. Eq. (4) shows that diffusion is not a heuristic (Zhu et al., 2003; Zhou et al., 2003): it is the solution of a convex problem trading off seed fidelity and Dirichlet energy on the token graph.

**Token graph and transition matrix.** Given token embeddings $Z = [z_1, \ldots, z_N]^\top \in \mathbb{R}^{N \times d}$, define a $k$-nearest-neighbor ($k$NN) graph under cosine similarity by assigning to each node $i$ a neighbor set $\mathcal{N}_k(i)$. A sparse *random-walk*

*(transition) matrix* (Von Luxburg, 2007) $W \in \mathbb{R}^{N \times N}$ is obtained by softmax normalization over outgoing edges,

$$W_{ij} = \begin{cases} \dfrac{\exp\left(\tau \sin(z_i, z_j)\right)}{\sum_{j' \in \mathcal{N}_k(i)} \exp\left(\tau \sin(z_i, z_{j'})\right)}, & j \in \mathcal{N}_k(i), \\ 0, & \text{otherwise,} \end{cases} \tag{5}$$

where $\text{sim}(z_i, z_j) = \frac{z_i^\top z_j}{\|z_i\|_2 \|z_j\|_2}$ and $\tau > 0$ is a temperature. By construction, $W$ is nonnegative and row-stochastic ($W\mathbf{1} = \mathbf{1}$), and thus admits the standard interpretation of one-step transition probabilities on the token graph. This matrix serves as the propagation operator in Eq. (2). [1]

## 3.2. Why Diffuse on a Feature-Similarity Graph Rather Than Raw Attention

Although attention maps provide a convenient training-free saliency cue, we do *not* use raw patch↔patch self-attention as the structural graph for diffusion. The reason is that in CLIP-style ViTs, raw self-attention is often not a reliable semantic affinity: it can connect tokens across inconsistent regions and does not preserve region coherence, as discussed in SCLIP (Wang et al., 2024a). Moreover, SC-CLIP (Bai et al., 2025) shows that anomaly tokens can emerge in deep layers and absorb attention mass from normal tokens, weakening spatial awareness and inducing feature homogenization. Therefore, using raw attention as a diffusion operator would propagate these artifacts, causing energy leakage to unrelated regions and reducing region-level discriminability under high compression. Accordingly, instead of diffusing on raw patch-to-patch attention links, we construct a $k$NN graph from token feature similarity and perform diffusion on this similarity-induced graph to preserve region coherence.

## 4. Method

Building on the above analysis, we propose SpecFlow, an efficient visual token pruning framework designed for high-performance vision-language modeling. SpecFlow leverages HeatFlow and adaptive quadtree allocation to better preserve the holistic contextual information and structural coherence of images. To avoid information loss, pruned tokens are compressed into compact coreset-style sink tokens to ensure statistical conservation.

### 4.1. HeatFlow: CLS-seeded energy diffusion

We instantiate the diffusion in Eq. (2) on the $k$NN graph in Eq. (5) to obtain a structure-aware token energy used for pruning. Algorithm 1 summarizes the overall procedure. We initialize the token energy from the attention distribution

---

[1] We use a symmetrized $k$NN edge set (i.e., include $(i, j)$ if $j \in \mathcal{N}_k(i)$ or $i \in \mathcal{N}_k(j)$), which makes the underlying affinity symmetric under cosine similarity, while keeping the graph sparse.

---

**Algorithm 1** CLS-seeded heat flow on a $k$NN token graph

---

**input** self-attention $A$, token embeddings $Z$
**output** diffused energy $E$

1: **for** $b = 1$ **to** $B$ **do**
2:     **Seed from CLS attention.** Select heads $\mathcal{H}$ and set
        $s_i \leftarrow \frac{1}{|\mathcal{H}|} \sum_{h \in \mathcal{H}} A_{b,h,0,i}$ for $i = 1, \dots, N$
3:     $s \leftarrow s/(\mathbf{1}^\top s + \varepsilon)$
4:     **Construct token transition matrix.** Form the row-stochastic matrix $W$ from $\{z_{b,i}\}_{i=1}^N$ by Eq. (5)
5:     $e \leftarrow s$
6:     **for** $t = 1$ **to** $T_{\text{diff}}$ **do**
7:         $e \leftarrow (1 - \alpha)s + \alpha W^\top e$
8:         $e \leftarrow e/(\mathbf{1}^\top e + \varepsilon)$
9:     **end for**
10:    $E_b \leftarrow e$
11: **end for**
12: **return** $E$

---

**Algorithm 2** Quadtree token pruning with energy allocation

---

**input** grid tokens $Z$ (size $H \times W$), diffused energy $e^\star$, budget $K$
**output** selected tokens $\tilde{Z}$

1: **for** $b = 1$ **to** $B$ **do**
2:     $E \leftarrow \text{Reshape}(e_b^\star, H, W)$
3:     **Quadtree splitting.** Starting from $(0, H, 0, W)$, recursively split a crop $c$ into four quadrants whenever $\text{split}(c)$ holds (Eq. (8)); denote the resulting leaf crops by $\mathcal{C}$
4:     **Energy-guided allocation.** For each leaf crop $c \in \mathcal{C}$, compute its energy mass $S(c)$ from $E$ as in Eq. (9) and initialize an integer token quota $q_c$ by Eq. (13); adjust $\{q_c\}$ to satisfy $\sum_{c \in \mathcal{C}} q_c = K$
5:     **Token selection.** $\mathcal{S} \leftarrow \bigcup_{c \in \mathcal{C}} \text{TopK}(E[c], q_c)$ and $\tilde{Z}_b \leftarrow Z_b[\mathcal{S}]$
6:     optionally: append sink tokens (mean and residual) from $Z_b[\bar{\mathcal{S}}]$ (Sec. 4.3)
7: **end for**
8: **return** $\tilde{Z}$

---

of the visual `[CLS]` token. In transformer-based vision encoders, `[CLS]` serves as a global aggregation token (Dosovitskiy, 2020; Radford et al., 2021) whose representation is optimized to summarize image-level semantics. Therefore, the outgoing attention weights from `[CLS]` provide a lightweight, training-free cue of which visual tokens contribute most to the global representation. Compared to local heuristics or post-hoc gradients, this signal is readily available during the forward pass and is naturally aligned with the model's internal routing of information. Formally, let $A \in \mathbb{R}^{(N+1) \times (N+1)}$ be the self-attention matrix at a chosen layer/head, where index 0 corresponds to `[CLS]` and indices $1, \dots, N$ correspond to visual tokens. We define the initial energy as

$$e_i^{(0)} = A_{0,i}, \quad i = 1, \dots, N, \qquad (6)$$

optionally aggregating multiple heads to reduce head-specific noise. This initialization highlights globally relevant regions while remaining computationally negligible, making it a suitable seed for the subsequent graph diffusion in Eq. (2). Starting from $e^{(0)}$, we run a small number of iterations of Eq. (2) to obtain $e^{(T)}$, and use it as the final energy $e^\star$ after normalization. Since $W$ is $k$-sparse, the update costs $\mathcal{O}(Nk)$ per iteration. We $\ell_1$-normalize $e$ after each iteration only for numerical stability, since $W$ is row-stochastic and $e^{(0)}$ is normalized, the diffusion preserves mass in exact arithmetic (Appendix A.1).

**4.2. Quadtree token pruning with energy allocation**

Given the diffused token energy $e^\star \in \mathbb{R}^N$ from Sec. 4.1, we select a compact subset of visual tokens under a budget $K$ while preserving spatial coverage. Since tokens are arranged on an $H \times W$ grid, we first reshape the energy into a 2D map $E \in \mathbb{R}^{H \times W}$ and then perform adaptive quadtree partitioning (Samet, 1984). Intuitively, we split regions with high energy variation into finer crops and keep low-variation regions coarse, enabling multi-scale selection. Algorithm 2 summarizes the overall pruning and allocation procedure.

**Energy map.** Let $E = \text{Reshape}(e^\star, H, W)$ denote the energy map aligned with the token grid. For any rectangular crop $c = (r_0, r_1, t_0, t_1)$, we use $E[c]$ to denote the set of energies inside $c$ and $|c| = (r_1 - r_0)(t_1 - t_0)$ its area.

**Quadtree splitting.** Starting from the full region $c_{\text{root}} = (0, H, 0, W)$, we recursively split a crop into four quadrants if it is (i) sufficiently large and (ii) non-uniform in energy. Concretely, we compute the crop mean and standard deviation

$$\mu(c) = \frac{1}{|c|} \sum_{(r,t) \in c} E_{r,t},$$
$$\sigma(c) = \sqrt{\frac{1}{|c|} \sum_{(r,t) \in c} \left(E_{r,t} - \mu(c)\right)^2}. \qquad (7)$$

and apply the split criterion

$$\text{split}(c) = \big(h(c) \geq 2m\big) \wedge \big(w(c) \geq 2m\big) \wedge \big(\sigma(c) > \delta\big), \qquad (8)$$

where $h(c) = r_1 - r_0$ and $w(c) = t_1 - t_0$ are the crop height/width, $m$ is the minimum crop size, and $\delta$ controls the sensitivity to energy variation. If $\text{split}(c)$ is true, we partition $c$ at midpoints $r_m = \lfloor (r_0 + r_1)/2 \rfloor$ and $t_m = \lfloor (t_0 + t_1)/2 \rfloor$ to obtain four children crops; otherwise $c$ becomes a leaf. We denote the set of all leaf crops by $\mathcal{C}$, which forms a disjoint cover of the grid.

**Energy-guided budget allocation.** Token budget is distributed across leaf crops in proportion to their energy mass. For each $c \in \mathcal{C}$, define

$$M(c) = \sum_{(r,t) \in c} E_{r,t}, \qquad (9)$$

and let $|c|$ denote the number of tokens in crop $c$. A fractional quota is given by

$$\hat{q}_c = K \cdot \frac{M(c)}{\sum_{c' \in \mathcal{C}} M(c')}. \qquad (10)$$

This proportional rule admits an optimization interpretation:

**Proposition 4.1** (Energy-proportional allocation as proportional fairness). *Assume $M(c) > 0$ for all $c \in \mathcal{C}$. The fractional quota $\{\hat{q}_c\}_{c \in \mathcal{C}}$ defined by*

$$\hat{q}_c = K \cdot \frac{M(c)}{\sum_{c' \in \mathcal{C}} M(c')} \qquad (11)$$

*is the unique optimizer of the concave program (Kelly et al., 1998)*

$$\max_{\{q_c > 0\}} \sum_{c \in \mathcal{C}} M(c) \log q_c \quad s.t. \quad \sum_{c \in \mathcal{C}} q_c = K. \qquad (12)$$

We provide the proof and integer rounding details in Appendix A.2.

An initial integer allocation is obtained by flooring and clipping to the feasible range,

$$q_c \leftarrow \min(|c|, \lfloor \hat{q}_c \rfloor), \qquad (13)$$

which allows $q_c = 0$ when the budget is smaller than the number of crops. Since flooring and clipping imply $\sum_{c \in \mathcal{C}} q_c \leq K$, the remaining budget $K - \sum_c q_c$ is distributed to crops in descending order of $S(c)$ among those with $q_c < |c|$, without exceeding $|c|$, until $\sum_{c \in \mathcal{C}} q_c = K$. Overall, the procedure enforces the global budget while maintaining $0 \leq q_c \leq |c|$.

**Token selection within each crop.** Within each crop $c$, the $q_c$ highest-energy tokens are retained:

$$\mathcal{S} = \bigcup_{c \in \mathcal{C}} \mathrm{TopK}(E[c], q_c), \qquad |\mathcal{S}| = K. \qquad (14)$$

The pruned token sequence is then $\tilde{Z} = Z[\mathcal{S}]$, where indices in $\mathcal{S}$ follow the original token order.

**Discussion.** The quadtree split enforces spatial adaptivity: salient regions (high-variation energy) are represented at a finer granularity, while background regions remain coarse. Coupled with energy-proportional allocation, this yields an efficient subset that preserves both semantic relevance (via $E$) and coverage (via the multi-scale partition).

## 4.3. Sink tokens for preserving pruned information

Quadtree pruning keeps a budgeted subset of tokens $\mathcal{S}$ but discards the complement $\bar{\mathcal{S}}$. While tokens in $\bar{\mathcal{S}}$ are assigned lower energy, they may still contain complementary context (e.g., background cues or fine-grained attributes). To mitigate information loss with negligible overhead, we compress the discarded set into a small number of additional *sink* tokens and feed them together with the selected tokens to the decoder. Let $Z \in \mathbb{R}^{N \times d}$ denote the token features, $\tilde{Z} = Z[\mathcal{S}]$ the selected tokens, and $Z_\mathrm{p} = Z[\bar{\mathcal{S}}]$ the pruned tokens. We summarize $Z_\mathrm{p}$ using two coreset-style sinks (Feldman & Langberg, 2011):

$$u_\mathrm{mean} = \frac{1}{|\bar{\mathcal{S}}|} \sum_{i \in \bar{\mathcal{S}}} Z_i, \qquad u_\mathrm{res} = Z_{\arg\max_{i \in \bar{\mathcal{S}}} \|Z_i - u_\mathrm{mean}\|_2}. \qquad (15)$$

The mean sink captures the overall context of discarded tokens, while the residual sink promotes diversity by retaining a representative token that is poorly explained by the mean. We append the sink tokens to the selected tokens to form the final visual token sequence

$$Z' = [\tilde{Z}; u_\mathrm{mean}; u_\mathrm{res}], \qquad (16)$$

which is passed to the language model.

## 4.4. Computational Complexity

We express inference cost in FLOPs and concentrate on visual tokens, as the text prompt is usually much shorter than the visual sequence (i.e., $T \ll N$). Let $N$ denote the number of visual tokens produced by the encoder and $K$ the number kept after pruning. Denote the hidden size by $h$ and the FFN intermediate size by $d$. Coefficients $a, b, c$ are implementation-dependent constants.

**Prefill FLOPs.** During prefill, the per-layer FLOPs as a function of the attended visual length $n$ can be modeled as

$$\Psi_\mathrm{pre}(n) := a h n^2 + n(bh^2 + chd), \qquad (17)$$

where the first term corresponds to attention and the latter terms capture per-token projections and the FFN. Pruning replaces $n = N$ with $n = K$, hence the relative prefill saving is

$$\Delta_\mathrm{pre} := 1 - \frac{\Psi_\mathrm{pre}(K)}{\Psi_\mathrm{pre}(N)} = 1 - \frac{ahK^2 + K(bh^2 + chd)}{ahN^2 + N(bh^2 + chd)}. \qquad (18)$$

Let $K = (1-R)N$. When attention dominates (typical for large $N$), the $n^2$ term governs and the reduction simplifies to

$$\Delta_\mathrm{pre} \approx 1 - \left(\frac{K}{N}\right)^2 = 1 - (1-R)^2 = 2R - R^2. \qquad (19)$$

highlighting that prefill gains are superlinear in the pruning ratio $R$ under this regime.

*Table 1.* **Comparison of pruning methods on image-understanding benchmarks.** We report the accuracy and average performance at varying pruning ratios. The best performance is marked in **red**.

| Methods | GQA | MMB | MMB$_{CN}$ | MME | POPE | SQA | VQA$_{V2}$ | VQA$_{Text}$ | VizWiz | Average |
|---|---|---|---|---|---|---|---|---|---|---|
| Upper Bound, 576 Tokens | 61.9 | 64.7 | 58.1 | 1862 | 85.9 | 69.5 | 78.4 | 58.2 | 50.0 | 100% |
| LLaVA-1.5 (Liu et al., 2024a) 7B | | | | | *Retain 192 Tokens* (↓ **66.7%**) | | | | | |
| ToMe (Bolya et al., 2022) (ICLR23) | 54.3 | 60.5 | - | 1563 | 72.4 | 65.2 | 68.0 | 52.1 | - | 88.5% |
| FastV (Chen et al., 2024b) (ECCV24) | 52.7 | 61.2 | 57.0 | 1612 | 64.8 | 67.3 | 67.1 | 52.5 | 50.8 | 90.5% |
| LLaVA-PruMerge (Shang et al., 2025) (ICCV25) | 54.3 | 59.6 | 52.9 | 1632 | 71.3 | 67.9 | 70.6 | 54.3 | 50.1 | 91.4% |
| PDrop (Xing et al., 2024) (CVPR25) | 57.1 | 63.2 | 56.8 | 1766 | 82.3 | 68.8 | 75.1 | 56.1 | 51.1 | 96.7% |
| HiRED (Arif et al., 2025) (AAAI25) | 58.7 | 62.8 | 54.7 | 1737 | 82.8 | 68.4 | 74.9 | 47.4 | 50.1 | 94.6% |
| VisionZip (Yang et al., 2025) (CVPR25) | **59.3** | 64.5 | 57.3 | 1767 | 86.4 | 68.9 | **76.8** | 57.3 | **51.6** | 98.1% |
| SparseVLM (Zhang et al., 2024b) (ICML25) | 57.6 | 62.5 | 53.7 | 1721 | 83.6 | 69.1 | 75.6 | 56.1 | 50.5 | 96.1% |
| DART (Wen et al., 2025) (EMNLP25) | 58.9 | 63.6 | **57.0** | **1856** | 82.8 | 69.8 | 76.7 | 57.4 | 51.1 | 98.5% |
| HoloV (Zou et al., 2025) (NeurIPS25) | 58.6 | 64.6 | 56.9 | 1793 | 85.6 | 69.1 | 76.1 | 55.8 | 51.4 | 98.2% |
| SpecFlow (Ours) | 58.3 | **65.8** | 56.1 | 1827 | **85.8** | **69.7** | **76.4** | **57.9** | 50.5 | **98.7%** |
| LLaVA-1.5 (Liu et al., 2024a) 7B | | | | | *Retain 128 Tokens* (↓ **77.8%**) | | | | | |
| ToMe (Bolya et al., 2022) (ICLR23) | 52.4 | 53.3 | - | 1343 | 62.8 | 59.6 | 63.0 | 49.1 | - | 80.4% |
| FastV (Chen et al., 2024b) (ECCV24) | 49.6 | 56.1 | 56.4 | 1490 | 59.6 | 60.2 | 61.8 | 50.6 | 51.3 | 85.4% |
| LLaVA-PruMerge (Shang et al., 2025) (ICCV25) | 53.3 | 58.1 | 51.7 | 1554 | 67.2 | 67.1 | 68.8 | 54.3 | 50.3 | 89.4% |
| PDrop (Xing et al., 2024) (CVPR25) | 56.0 | 61.1 | 56.6 | 1644 | 82.3 | 68.3 | 72.9 | 55.1 | 51.0 | 94.9% |
| HiRED (Arif et al., 2025) (AAAI25) | 57.2 | 61.5 | 53.6 | 1710 | 79.8 | 68.1 | 73.4 | 46.1 | 51.3 | 93.1% |
| VisionZip (Yang et al., 2025) (CVPR25) | 57.6 | 63.4 | 56.7 | 1768 | 84.7 | 68.8 | 75.6 | 56.8 | 52.0 | 97.2% |
| SparseVLM (Zhang et al., 2024b) (ICML25) | 56.0 | 60.0 | 51.1 | 1696 | 80.5 | 67.1 | 73.8 | 54.9 | 51.4 | 93.8% |
| DART (Wen et al., 2025) (EMNLP25) | **57.9** | 63.2 | **57.0** | **1845** | 80.1 | 69.1 | **75.9** | 56.4 | 51.7 | 97.5% |
| HoloV (Zou et al., 2025)(NeurIPS25) | 57.4 | 62.8 | 55.0 | 1768 | 83.2 | 69.1 | 74.8 | 55.7 | 52.3 | 96.8% |
| SpecFlow (Ours) | 57.6 | **64.3** | 55.9 | 1794 | **84.9** | **69.9** | 75.3 | **56.8** | 51.1 | **97.8%** |
| LLaVA-1.5 (Liu et al., 2024a) 7B | | | | | *Retain 64 Tokens* (↓ **88.9%**) | | | | | |
| ToMe (Bolya et al., 2022) (ICLR23) | 48.6 | 43.7 | - | 1138 | 52.5 | 50.0 | 57.1 | 45.3 | - | 70.1% |
| FastV (Chen et al., 2024b) (ECCV24) | 46.1 | 48.0 | 52.7 | 1256 | 48.0 | 51.1 | 55.0 | 47.8 | 50.8 | 76.7% |
| LLaVA-PruMerge (Shang et al., 2025) (ICCV25) | 51.9 | 55.3 | 49.1 | 1549 | 65.3 | 68.1 | 67.4 | 54.0 | 50.1 | 87.7% |
| PDrop (Xing et al., 2024) (CVPR25) | 41.9 | 33.3 | 50.5 | 1092 | 55.9 | 68.6 | 69.2 | 45.9 | 50.7 | 77.5% |
| HiRED (Arif et al., 2025) (AAAI25) | 54.6 | 60.2 | 51.4 | 1599 | 73.6 | 68.2 | 69.7 | 44.2 | 50.2 | 89.4% |
| VisionZip (Yang et al., 2025) (CVPR25) | 55.1 | 60.1 | **55.4** | 1690 | 77.0 | 69.0 | 72.4 | **55.5** | 52.9 | 94.5% |
| SparseVLM (Zhang et al., 2024b) (ICML25) | 52.7 | 56.2 | 46.1 | 1505 | 75.1 | 62.2 | 68.2 | 51.8 | 50.1 | 87.3% |
| DART (Wen et al., 2025) (EMNLP25) | **55.9** | 60.6 | 53.2 | **1765** | 73.9 | **69.8** | 72.4 | 54.4 | 51.6 | 93.9% |
| HoloV (Zou et al., 2025)(NeurIPS25) | 55.1 | 61.3 | 53.7 | 1728 | 80.3 | 69.5 | 72.2 | 54.2 | **53.1** | 94.9% |
| SpecFlow (Ours) | 55.3 | **63.7** | 54.3 | 1713 | **80.5** | 69.7 | **73.7** | 54.9 | 52.4 | **95.6%** |

**Decode FLOPs.** With KV caching, each decoding step scales approximately linearly with the context length:

$$\Psi_{dec}(n) := bh^2 + n(bh + chd). \quad (20)$$

Replacing $N$ by $K$ gives

$$\Delta_{dec} := 1 - \frac{\Psi_{dec}(K)}{\Psi_{dec}(N)} = 1 - \frac{bh^2 + K(bh+chd)}{bh^2 + N(bh+chd)} \approx R. \quad (21)$$

since the constant term $bh^2$ does not shrink with pruning.

**Cost of pruning.** Pruning is performed once per input; in particular, the $k$NN graph is computed only once before decoder-layer processing. The overhead comprises $k$NN graph construction (worst-case $\mathcal{O}(N^2h)$ FLOPs), sparse diffusion $\mathcal{O}(T_{diff}Nk)$, and quadtree splitting/TopK selection near-linear in $N$ (i.e., $\tilde{\mathcal{O}}(N)$, or $\mathcal{O}(DN)$ with depth $D$). As these costs are not incurred per decoder layer, they are amortized by the per-layer savings from reducing the attended length from $N$ to $K$, especially during prefill. A more detailed quantitative breakdown of the pruning overhead, end-to-end latency, and memory footprint, together with concrete numerical instantiations of the FLOPs formulas above, is provided in Appendix C.

## 5. Experiments

### 5.1. Experimental Setup

**Benchmarks.** We validate our approach on a diverse suite of established benchmarks for visual understanding. For image understanding task, we consider ten datasets, including GQA (Hudson & Manning, 2019), MMBench (MMB) and MMB-CN (Liu et al., 2024d), MME (Fu et al., 2023), POPE (Li et al., 2023), VizWiz (Bigham et al., 2010), ScienceQA (SQA) (Lu et al., 2022), VQA v2 (VQA$_{V2}$) (Goyal et al., 2017) and TextVQA (VQA$_{Text}$) (Singh et al., 2019). For video understanding task, we additionally evaluate on MSVD-QA and MSRVTT-QA (Xu et al., 2017). We follow the official evaluation protocols and default settings for each benchmark. Further benchmark details are provided in Appendix B.1.

### 5.2. Main Results

**Image Understanding Tasks.** Table 1 summarizes the main results on nine image-understanding benchmarks under different token budgets. Overall, SpecFlow consis-

tently achieves a strong accuracy–efficiency trade-off across datasets and compression regimes. At a 66.7% token compression ratio, SpecFlow attains the best average relative performance (98.7%), incurring only a 1.3% drop from the uncompressed full-token baseline (100%), while outperforming recent pruning baselines such as VisionZip, Sparse-VLM, and HoloV, and remaining competitive on individual benchmarks. Notably, on MMBench and SQA, SpecFlow even surpasses the unpruned baseline. As the token budget becomes more stringent, SpecFlow degrades gracefully and continues to deliver the best averages at 77.8% and 88.9% token compression ratios (97.8% and 95.6%, respectively).

**Results on LLaVA-NeXT.** For a more thorough evaluation, we further benchmark SpecFlow on LLaVA-NeXT across the same suite of datasets and compare against current state-of-the-art training-free pruning methods. Since LLaVA-NeXT adopts a revised image encoding pipeline, the number of visual tokens varies across inputs. To ensure a consistent evaluation protocol, we fix the token budget at 320 (pruned from up to 2880 raw tokens). As reported in Table 2, SpecFlow delivers the strongest overall performance, while retaining 95.8% of the unpruned baseline performance with an 88.9% reduction in visual tokens.

**Video Understanding Tasks.** We evaluate our method on two widely used video question-answering benchmarks. Following the protocol in (Chen et al., 2024b; Zhang et al., 2024b), we report results on the first 1,000 samples of each benchmark and use the Video-ChatGPT (Maaz et al., 2024) score as the primary metric. In Table 3, we treat Video-LLaVA with 2,048 video tokens as the unpruned upper bound, normalized to an average accuracy of 100.0% and a score difference of +0.00. For a fair comparison, all pruning methods retain 455 visual tokens (77.8% pruning ratio). Under this setting, SpecFlow preserves performance close to the unpruned baseline and consistently outperforms FastV, SparseVLM, and HoloV. These results indicate that SpecFlow remains effective for video inputs with temporal dynamics, producing accurate answers while substantially reducing token usage.

**Results on Qwen2.5-VL.** To assess architectural generality beyond the LLaVA family, we further evaluate SpecFlow on Qwen2.5-VL-7B, which differs from LLaVA in both the visual stack and the multimodal projector. As reported in Table 4, SpecFlow consistently outperforms HoloV across all three pruning ratios, with the margin growing under more aggressive compression (e.g., 89.4% vs. 87.3% relative accuracy at 88.9% pruning). This indicates that the proposed mechanism, which combines graph diffusion with adaptive regional allocation and sink-based context preservation, generalizes well to different MLLM architectures and is not tied to a specific CLIP-style encoder or projector design.

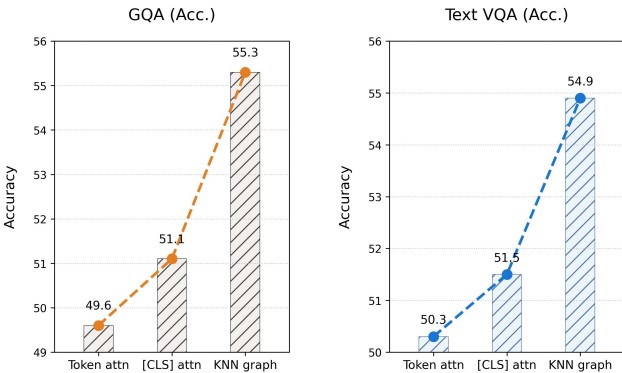

*Figure 2.* Ablation study on diffusion operators. Our KNN graph (built from token feature similarity) outperforms raw self-attention (Token attn) and `[CLS]`-initialized attention propagation (`[CLS]` attn).

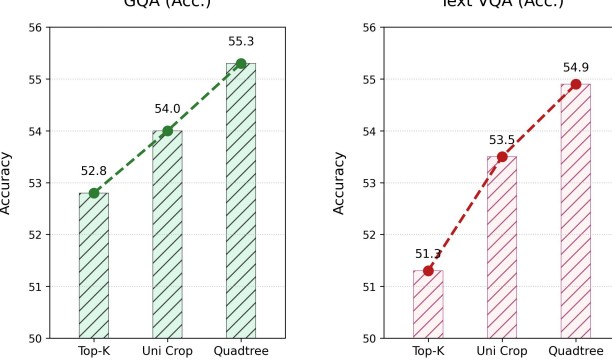

*Figure 3.* Ablation on pruning strategies. Our Quadtree outperforms Top-K (direct selection of top-K tokens) and Uni Crop (uniformly partitioned crops), validating adaptive spatial pruning.

### 5.3. Ablation Studies

**Ablation on Diffusion Operator.** We validate the design choice of constructing a KNN graph based on token feature similarity by comparing it against two variants: raw visual token self-attention (*Token attn*) and `[CLS]`-initialized attention propagation (`[CLS] attn`). As illustrated in Figure 3, our KNN graph-based diffusion consistently outperforms both baselines. Specifically, it achieves 55.3% on GQA and 54.9% on TextVQA, surpassing the *Token attn* baseline by 5.7% and 4.6%, respectively. These results verify that feature-similarity $k$NN graphs provide a robust structural basis for diffusion, effectively preserving region coherence while mitigating the noise and artifacts inherent in raw attention maps.

**Ablation on Adaptive Spatial Pruning Strategies.** To assess the effectiveness of our adaptive quadtree-based pruning, we compare it against two non-adaptive strategies: *Top-K*, which selects high-energy tokens without spatial constraints; and *Uni Crop*, which employs rigid uniform partitioning. As shown in the results, our quadtree approach yields superior performance, reaching 55.3% on GQA and 54.9% on TextVQA. Critically, our method ad-

*Table 2.* **Comparison of pruning methods on Video QA benchmarks.** We report the accuracy and average performance at 88.9% pruning ratios. The best performance is marked in red.

| Methods | GQA | MMB | MMB$_{CN}$ | MME | POPE | SQA | VQA$_{V2}$ | VQA$_{Text}$ | VizWiz | Average |
|---|---|---|---|---|---|---|---|---|---|---|
| Upper Bound, 2880 Tokens | 64.2 | 67.4 | 60.6 | 1851 | 86.5 | 70.1 | 81.8 | 64.9 | 57.6 | 100% |
| LLaVA-NeXT (Liu et al., 2024b) 7B | | | | *Retain 320 Tokens* (↓**88.9%**) | | | | | | |
| FastV (Chen et al., 2024b) (ECCV24) | 55.9 | 61.6 | 51.9 | 1661 | 71.7 | 62.8 | 71.9 | 55.7 | 53.1 | 88.0% |
| LLaVA-PruMerge (Shang et al., 2025) (ICCV25) | 53.6 | 61.3 | 55.3 | 1534 | 60.8 | 66.4 | 69.7 | 50.6 | 54.0 | 85.6% |
| PDrop (Xing et al., 2024) (CVPR25) | 56.4 | 63.4 | 56.2 | 1663 | 77.6 | 67.5 | 73.5 | 54.4 | 54.1 | 90.9% |
| FasterVLM (Zhang et al., 2024a) (ICCV25) | 56.9 | 61.6 | 53.5 | 1701 | 83.6 | 66.5 | 74.0 | 56.5 | 52.6 | 91.1% |
| HiRED (Arif et al., 2025) (AAAI25) | 59.3 | 64.2 | 55.9 | 1690 | 83.3 | 66.7 | 75.7 | 58.8 | 54.2 | 93.3% |
| SparseVLM (Zhang et al., 2024b) (ICML25) | 56.1 | 60.6 | 54.5 | 1533 | 82.4 | 66.1 | 71.5 | 58.4 | 52.0 | 89.7% |
| DART (Wen et al., 2025) (EMNLP25) | 61.7 | 65.3 | **58.2** | 1710 | 84.1 | 68.4 | 79.1 | 58.7 | **56.1** | 93.9% |
| HoloV (Zou et al., 2025) (NeurIPS25) | 61.7 | 65.3 | 57.5 | **1738** | 83.9 | **68.9** | 79.5 | 58.7 | 55.3 | 95.6% |
| SpecFlow (Ours) | **62.5** | **66.7** | 56.8 | 1707 | **85.0** | 68.6 | **80.1** | **59.5** | 54.2 | **95.8%** |

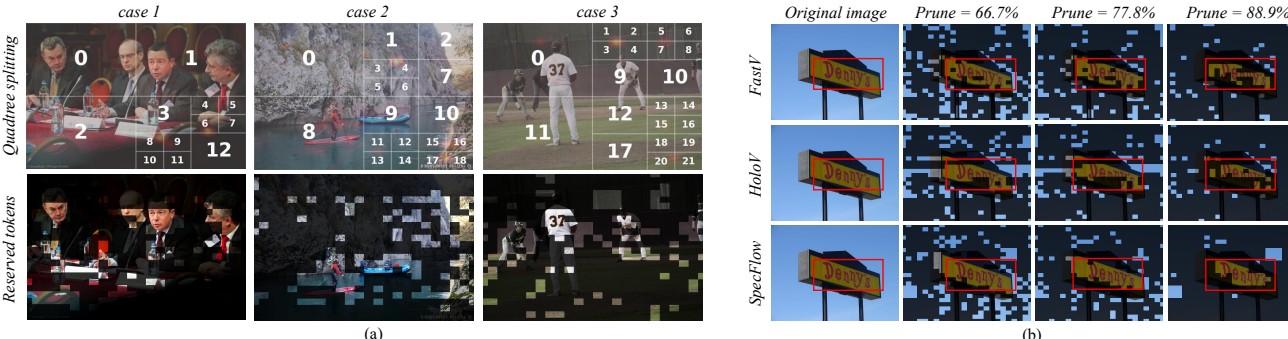

*Figure 4.* Qualitative visualization of token selection and adaptive splitting. (a) Visualization of our energy-based quadtree splitting, showing the induced partitions and the final retained-token layouts across representative cases (b) Comparison of retained visual tokens at different pruning ratios for SpecFlow versus FastV and HoloV.

*Table 3.* Video QA Evaluations of different methods with 455 visual tokens retained.

| Methods | MSVD-QA | | MSRVT-QA | | Average | |
|---|---|---|---|---|---|---|
| | Acc. | Score | Acc. | Score | Acc. | Score |
| Video-LLaVA 7B | 70.8 | 3.93 | 57.5 | 3.55 | 64.2 | 3.74 |
| FastV (ECCV24) | 68.2 | 3.75 | 54.1 | 3.42 | 61.2 | 3.59 |
| SparseVLM (ICML25) | 69.4 | 3.89 | 54.8 | 3.42 | 62.1 | 3.66 |
| HoloV (NeurIPS25) | 69.3 | 3.90 | 56.2 | 3.50 | 62.8 | 3.70 |
| SpecFlow (Ours) | **70.3** | **3.94** | **56.5** | **3.50** | **63.4** | **3.72** |

*Table 4.* SpecFlow Generalization on Qwen2.5-VL.

| Methods | MME | POPE | SQA | VQA$_{Text}$ | Avg. |
|---|---|---|---|---|---|
| Upper Bound | 2304 | 86.1 | 84.7 | 84.8 | 100.0% |
| Qwen2.5-VL-7B | | *Retain 192 Tokens* (↓66.7%) | | | |
| HoloV (NeurIPS25) | 2066 | 85.0 | 79.1 | 77.3 | 93.2% |
| SpecFlow (Ours) | **2102** | **85.2** | **80.0** | **79.2** | **94.5%** |
| Qwen2.5-VL-7B | | *Retain 128 Tokens* (↓77.8%) | | | |
| HoloV (NeurIPS25) | 2029 | 81.5 | 79.1 | 69.2 | 89.4% |
| SpecFlow (Ours) | **2051** | **83.7** | **79.9** | **73.8** | **91.9%** |
| Qwen2.5-VL-7B | | *Retain 64 Tokens* (↓88.9%) | | | |
| HoloV (NeurIPS25) | 1998 | 80.7 | 79.5 | 63.6 | 87.3% |
| SpecFlow (Ours) | **2015** | **81.1** | **79.7** | **69.4** | **89.4%** |

dresses the limitations of the baselines: while *Top-K* often results in spatial fragmentation and *Uni Crop* inefficiently allocates budget to low-importance backgrounds, our quadtree method adaptively refines granularity based on energy variation. This mechanism creates finer partitions in semantically salient regions to preserve detail, and coarser ones in uniform backgrounds to reduce redundancy. This design strikes an optimal balance between semantic preservation and spatial coverage, translating into significant performance gains.

## 5.4. Discussion

SpecFlow's advantage stems from coupling a structure-aware importance signal with spatially balanced token retention. We perform energy diffusion on the token graph so that semantically similar tokens can propagate importance to each other, which mitigates the fragmentation of coherent objects under aggressive pruning and prevents a small set of peak tokens from dominating the budget. Meanwhile, the diffused energy remains sufficiently discriminative, avoiding the overly smooth behavior that text-vision attention can exhibit (Zhang et al., 2025; Zou et al., 2025), and thus better separates salient from non-salient tokens, as evidenced

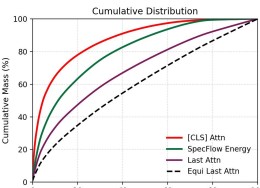

*Figure 5.* Cumulative mass (sorted) over visual token proportion for different importance measures.

by the cumulative mass distribution in Figure 5. Building on this importance signal, we further introduce an energy-based quadtree splitting strategy that adaptively partitions the image into regions of varying visual complexity and assigns a per-region token quota. As illustrated in Figure 4(a), this regional budgeting preserves object integrity while increasing spatial coverage, producing retained-token layouts that capture key structures across the scene instead of collapsing the budget onto a small area. Consequently, under extreme pruning ratios, SpecFlow maintains substantially better subject completeness than existing methods, as shown in Figure 4(b).

## 6. Conclusion

We presented SpecFlow, a training-free framework for efficient VLM inference that replaces destructive token pruning with conservative token condensation. SpecFlow computes a stable importance field via spectral heat diffusion on a $k$NN token graph, enforces spatial coverage through adaptive quadtree budgeting, and summarizes removed tokens with lightweight sink tokens to preserve context and diversity. Experiments on multiple image and video benchmarks show that SpecFlow achieves substantial visual token reduction with minimal accuracy loss, outperforming strong training-free baselines and remaining compatible with FlashAttention.

## Acknowledgments

This work was supported by the Open Fund of National Key Laboratory of Deep Space Exploration (NKDSEL2025008).

## Impact Statement

This paper presents SpecFlow, a training-free framework for efficient inference in vision-language models (VLMs) via conservative token condensation. By selecting a smaller set of region-coherent visual tokens and aggregating pruned information into lightweight sink tokens, SpecFlow can reduce the computational and memory cost of VLM inference, especially in the prefill stage. These efficiency gains may lower latency and monetary cost per query and reduce energy consumption, potentially enabling broader access to real-time and on-device VLM applications such as assistive interfaces, interactive education, and robotics.

However, improved efficiency can also lower the barrier to deploying VLMs at scale. When combined with powerful generative or retrieval systems, faster VLM inference could facilitate high-volume uses that raise ethical concerns, including privacy-invasive monitoring, large-scale content analysis, or the amplification of misleading or harmful multimodal content. While SpecFlow does not introduce new model capabilities or new training data by itself, it can increase throughput and thus the scale at which downstream systems are used. We therefore recommend that deployments preserve and strengthen existing safety measures, including content filtering, rate limiting, logging/auditing, and compliance with applicable privacy and data-protection regulations.

A further risk is that aggressive token budgets may degrade performance on some inputs (e.g., small objects, dense text, or rare visual concepts), which could be harmful in high-stakes settings. We recommend validating token budgets on the target domain, monitoring failure cases, and providing fallbacks (e.g., less compression, full-token inference, or human oversight) when uncertainty is high. Finally, SpecFlow inherits limitations and potential biases of the underlying VLMs and evaluation datasets; practitioners should consider domain-specific bias and robustness testing before deployment.

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

# A. Additional Theoretical Details

## A.1. HeatFlow Diffusion: Fixed Point, Mass Preservation, and Dirichlet View

This appendix provides proofs and auxiliary lemmas for Proposition 3.1. Throughout, $W \in \mathbb{R}^{N \times N}$ is the nonnegative row-stochastic transition matrix in Eq. (5) (i.e., $W\mathbf{1} = \mathbf{1}$), and $\alpha \in (0, 1)$ is the diffusion strength in Eq. (2).

### A.1 Contraction and unique fixed point (no symmetry required).    Define the restart-diffusion operator

$$\mathcal{T}(e) \triangleq (1 - \alpha)e^{(0)} + \alpha W^\top e. \tag{22}$$

**Lemma A.1** (Contraction and unique fixed point of Eq. (2)). *Assume $W \geq 0$ and $W\mathbf{1} = \mathbf{1}$ with $\alpha \in (0, 1)$. Then $\mathcal{T}$ in Eq. (22) is an $\alpha$-contraction under the $\ell_1$ norm:*

$$\|\mathcal{T}(e) - \mathcal{T}(e')\|_1 \leq \alpha \|e - e'\|_1, \quad \forall e, e' \in \mathbb{R}^N. \tag{23}$$

*Consequently, Eq. (2) admits a unique fixed point $e^\star$ satisfying*

$$(\mathbf{I} - \alpha W^\top)e^\star = (1 - \alpha)e^{(0)}, \tag{24}$$

*and the iterates converge linearly:* $\|e^{(t)} - e^\star\|_1 \leq \alpha^t \|e^{(0)} - e^\star\|_1$.

*Proof.* For any matrix $M$, $\|Mx\|_1 \leq \|M\|_1 \|x\|_1$ where $\|M\|_1 = \max_j \sum_i |M_{ij}|$ (Horn & Johnson, 2012). Since $W \geq 0$ and each row of $W$ sums to 1, we have

$$\|W^\top\|_1 = \max_j \sum_i (W^\top)_{ij} = \max_j \sum_i W_{ji} = \max_j \left( \sum_i W_{ji} \right) = 1.$$

Therefore,

$$\|\mathcal{T}(e) - \mathcal{T}(e')\|_1 = \alpha \|W^\top(e - e')\|_1 \leq \alpha \|W^\top\|_1 \|e - e'\|_1 = \alpha \|e - e'\|_1,$$

proving Eq. (23). By Banach's fixed-point theorem (Kreyszig, 1991), $\mathcal{T}$ has a unique fixed point $e^\star$. The fixed-point equation $\mathcal{T}(e^\star) = e^\star$ is equivalent to Eq. (24). The linear convergence bound follows directly from contraction. $\qquad\square$

### A.2 Mass preservation (justifies optional renormalization).

**Lemma A.2** (Mass preservation under restart diffusion). *Assume $W \geq 0$ and $W\mathbf{1} = \mathbf{1}$. If $\mathbf{1}^\top e^{(0)} = 1$ and $e^{(0)} \geq 0$, then for all $t \geq 0$,*

$$\mathbf{1}^\top e^{(t)} = 1, \qquad e^{(t)} \geq 0. \tag{25}$$

*Proof.* Nonnegativity follows by induction since $e^{(t+1)} = (1 - \alpha)e^{(0)} + \alpha W^\top e^{(t)}$ is a nonnegative combination of nonnegative vectors (as $W^\top \geq 0$). For mass, use $\mathbf{1}^\top W^\top = (W\mathbf{1})^\top = \mathbf{1}^\top$:

$$\mathbf{1}^\top e^{(t+1)} = (1 - \alpha)\mathbf{1}^\top e^{(0)} + \alpha \mathbf{1}^\top W^\top e^{(t)} = (1 - \alpha) \cdot 1 + \alpha \mathbf{1}^\top e^{(t)}.$$

If $\mathbf{1}^\top e^{(t)} = 1$, then $\mathbf{1}^\top e^{(t+1)} = 1$. Since it holds at $t = 0$, it holds for all $t$. $\qquad\square$

### A.3 Dirichlet view under symmetrized affinity (proof of Prop. 3.1).    Proposition 3.1 additionally assumes that $W$ is obtained by row-normalizing a symmetric affinity matrix. Concretely, define $\mathbf{S} \in \mathbb{R}^{N \times N}$ and $D = \text{diag}(\mathbf{S1})$ such that

$$\mathbf{S} = \mathbf{S}^\top \geq 0, \qquad D_{ii} > 0, \qquad W = D^{-1}\mathbf{S}. \tag{26}$$

This holds, for instance, when we use a symmetrized $KNN$ edge set and set $S_{ij} = \exp(\tau \, \text{sim}(z_i, z_j))$ on edges and 0 otherwise.

**Lemma A.3** (Dirichlet energy identity). *If $\mathbf{S} = \mathbf{S}^\top \geq 0$ and $D = \text{diag}(\mathbf{S1})$, then for any $f \in \mathbb{R}^N$,*

$$f^\top(D - \mathbf{S})f = \frac{1}{2} \sum_{i=1}^N \sum_{j=1}^N S_{ij}(f_i - f_j)^2. \tag{27}$$

*Proof.* Expand the RHS:

$$\sum_{i,j} S_{ij}(f_i - f_j)^2 = \sum_{i,j} S_{ij}(f_i^2 + f_j^2 - 2f_i f_j) = 2\sum_i \Big(\sum_j S_{ij}\Big)f_i^2 - 2\sum_{i,j} S_{ij}f_i f_j.$$

Since $\sum_j S_{ij} = D_{ii}$ and $\sum_{i,j} S_{ij}f_i f_j = f^\top \mathbf{S}f$, the RHS equals $2f^\top Df - 2f^\top \mathbf{S}f = 2f^\top(D - \mathbf{S})f$. Divide by 2 to obtain Eq. (27). $\qquad\square$

*Proof of Proposition 3.1.* We first show that the fixed point exists and is unique. Under Eq. (26), $W$ is row-stochastic, so Lemma A.1 applies and guarantees a unique fixed point $e^\star$ solving $(\mathbf{I} - \alpha W^\top)e^\star = (1 - \alpha)e^{(0)}$.

Next, define the objective in Proposition 3.1:

$$\mathcal{J}(e) = \frac{1}{2}\big\|D^{-1}e - D^{-1}e^{(0)}\big\|_D^2 + \frac{\alpha}{2(1 - \alpha)}(D^{-1}e)^\top(D - \mathbf{S})(D^{-1}e),$$

where $\|u\|_D^2 = u^\top D u$. Let $f = D^{-1}e$ and $f_0 = D^{-1}e^{(0)}$. Then $\mathcal{J}(e)$ can be rewritten as a function of $f$:

$$\mathcal{J}(e) \equiv \widetilde{\mathcal{J}}(f) = \frac{1}{2}\|f - f_0\|_D^2 + \frac{\alpha}{2(1 - \alpha)}f^\top(D - \mathbf{S})f. \tag{28}$$

Since $D \succ 0$ (diagonal with positive entries) and $(D - \mathbf{S}) \succeq 0$ (graph Laplacian), $\widetilde{\mathcal{J}}$ is strictly convex and thus admits a unique minimizer (Boyd & Vandenberghe, 2004).

Compute the gradient w.r.t. $f$:

$$\nabla_f \widetilde{\mathcal{J}}(f) = D(f - f_0) + \frac{\alpha}{1 - \alpha}(D - \mathbf{S})f.$$

Setting $\nabla_f \widetilde{\mathcal{J}}(f) = 0$ and multiplying by $(1 - \alpha)$ yields

$$(1 - \alpha)D(f - f_0) + \alpha(D - \mathbf{S})f = 0 \quad \Longleftrightarrow \quad (D - \alpha\mathbf{S})f = (1 - \alpha)Df_0.$$

Substituting $f = D^{-1}e$ and $f_0 = D^{-1}e^{(0)}$ gives

$$(D - \alpha\mathbf{S})D^{-1}e = (1 - \alpha)e^{(0)} \quad \Longleftrightarrow \quad (\mathbf{I} - \alpha\mathbf{S}D^{-1})e = (1 - \alpha)e^{(0)}.$$

Finally, since $W^\top = (D^{-1}\mathbf{S})^\top = \mathbf{S}D^{-1}$, we obtain

$$(\mathbf{I} - \alpha W^\top)e = (1 - \alpha)e^{(0)},$$

which matches the fixed-point equation. By uniqueness of the minimizer, the optimizer of $\mathcal{J}(e)$ is exactly $e^\star$. $\qquad\square$

### A.2. Quota Allocation and Integer Rounding for Quadtree Pruning

This appendix provides proof for Proposition 4.1 and a feasibility guarantee for the integer rounding procedure described in Sec. 4.2. Recall that $\mathcal{C}$ denotes the set of quadtree leaf crops, $M(c) = \sum_{(r,t)\in c} E_{r,t}$ is the energy mass, and the fractional quota is

$$\hat{q}_c = K \cdot \frac{M(c)}{\sum_{c'\in\mathcal{C}} M(c')}. \tag{29}$$

*Proof of Proposition 4.1.* Consider the concave optimization problem in Eq. (12) (Kelly et al., 1998):

$$\max_{\{q_c > 0\}} \sum_{c\in\mathcal{C}} M(c)\log q_c \quad \text{s.t.} \quad \sum_{c\in\mathcal{C}} q_c = K.$$

The Lagrangian is $\mathcal{L}(q, \lambda) = \sum_c M(c)\log q_c - \lambda\big(\sum_c q_c - K\big)$. Stationarity gives $\partial\mathcal{L}/\partial q_c = M(c)/q_c - \lambda = 0$, hence $q_c = M(c)/\lambda$ for all $c$. Imposing $\sum_c q_c = K$ yields $\lambda = \frac{\sum_c M(c)}{K}$, and thus $q_c = K \cdot \frac{M(c)}{\sum_{c'} M(c')} \equiv \hat{q}_c$. Since the objective is strictly concave over $q_c > 0$, the optimizer is unique. $\qquad\square$

**Feasible integer rounding with capacity constraints.** In Sec. 4.2, we form the initial integer quotas by flooring and clipping: $q_c^{(0)} \leftarrow \min(|c|, \lfloor \hat{q}_c \rfloor)$ (Eq. (13)), and then distribute the remaining budget to crops with available capacity until $\sum_c q_c = K$.

**Lemma A.4** (Feasibility of the floor+clip+fill rounding procedure). *Assume $0 < K \leq N$ and $\sum_{c \in \mathcal{C}} |c| = N$ (i.e., $\mathcal{C}$ is a disjoint cover of the $H \times W$ grid). Let $q_c^{(0)} = \min(|c|, \lfloor \hat{q}_c \rfloor)$, and let $R = K - \sum_{c \in \mathcal{C}} q_c^{(0)} \geq 0$. Consider any procedure that performs $R$ unit increments, each time choosing a crop $c$ with $q_c < |c|$ and setting $q_c \leftarrow q_c + 1$. Then the resulting integer quotas satisfy*

$$\sum_{c \in \mathcal{C}} q_c = K, \qquad 0 \leq q_c \leq |c| \ \ \forall c \in \mathcal{C}. \tag{30}$$

*In particular, the "descending $M(c)$" filling strategy in Sec. 4.2 is feasible.*

*Proof.* By construction, $0 \leq q_c^{(0)} \leq |c|$ and $\sum_c q_c^{(0)} \leq \sum_c \hat{q}_c = K$, hence $R \geq 0$. The total remaining capacity after the initial step is

$$\sum_c (|c| - q_c^{(0)}) \ \geq \ \sum_c |c| - \sum_c q_c^{(0)} \ = \ N - \sum_c q_c^{(0)} \ \geq \ K - \sum_c q_c^{(0)} \ = \ R,$$

where we used $K \leq N$. Therefore, there exists sufficient capacity to perform $R$ unit increments without violating $q_c \leq |c|$. Each increment increases $\sum_c q_c$ by 1 while maintaining $q_c \leq |c|$ by choice of $c$. After exactly $R$ increments, $\sum_c q_c = \sum_c q_c^{(0)} + R = K$. $\square$

### A.3. Properties of Sink Tokens (Mean + Residual)

This appendix provides basic properties of the sink construction in Eq. (15). Let $\bar{\mathcal{S}}$ denote the pruned index set, and assume $|\bar{\mathcal{S}}| > 0$. (If $|\bar{\mathcal{S}}| = 0$, sink tokens are omitted.)

**Lemma A.5** (Mean sink is the optimal 1-mean (least-squares prototype)). *The mean sink $u_{\mathrm{mean}} = \frac{1}{|\bar{\mathcal{S}}|} \sum_{i \in \bar{\mathcal{S}}} X_i$ is the unique minimizer of*

$$\min_{u \in \mathbb{R}^d} \sum_{i \in \bar{\mathcal{S}}} \|X_i - u\|_2^2. \tag{31}$$

*Proof.* Expand the objective: $\sum_i \|X_i - u\|_2^2 = \sum_i (\|X_i\|_2^2 - 2u^\top X_i + \|u\|_2^2)$. Differentiating w.r.t. $u$ gives $\nabla_u = -2\sum_i X_i + 2|\bar{\mathcal{S}}| u$. Setting $\nabla_u = 0$ yields $u = \frac{1}{|\bar{\mathcal{S}}|} \sum_i X_i = u_{\mathrm{mean}}$. Strict convexity in $u$ implies uniqueness. $\square$

**Lemma A.6** (Residual sink as a radius certificate). *Let $u_{\mathrm{res}} = X_{i^\star}$ where $i^\star = \arg\max_{i \in \bar{\mathcal{S}}} \|X_i - u_{\mathrm{mean}}\|_2$, and define the residual radius $r \triangleq \|u_{\mathrm{res}} - u_{\mathrm{mean}}\|_2$. Then for all $i \in \bar{\mathcal{S}}$, $\|X_i - u_{\mathrm{mean}}\|_2 \leq r$. Moreover, for any vector $v \in \mathbb{R}^d$,*

$$\max_{i \in \bar{\mathcal{S}}} |v^\top (X_i - u_{\mathrm{mean}})| \leq \|v\|_2 \, r. \tag{32}$$

*Proof.* The first claim follows directly from the definition of $i^\star$. For the second, Cauchy–Schwarz yields $|v^\top (X_i - u_{\mathrm{mean}})| \leq \|v\|_2 \|X_i - u_{\mathrm{mean}}\|_2 \leq \|v\|_2 \, r$, and taking the maximum over $i$ completes the proof. $\square$

## B. Detailed Experiment Settings

### B.1. Benchmarks

We evaluated our method on several widely used benchmarks for visual understanding. For image understanding, we considered nine datasets: GQA (Hudson & Manning, 2019); MMBench (MMB)(Liu et al., 2024d) and its Chinese split MMB-CN (Liu et al., 2024d); MME (Fu et al., 2023); POPE (Li et al., 2023); VizWiz (Bigham et al., 2010); SQA (ScienceQA) (Lu et al., 2022); VQA v2 (VQAV2) (Goyal et al., 2017); and TextVQA (VQAText) (Singh et al., 2019).

**GQA** (Hudson & Manning, 2019) GQA is a compositional visual question answering benchmark built around three core elements: images, structured scene annotations (scene graphs), and questions. Beyond the raw images, the dataset provides object-level information such as locations and attributes. Its questions are designed to probe not only visual recognition but also relational and compositional reasoning over the depicted scene.

**MMBench** (Liu et al., 2024d). MMBench is a multi-dimensional benchmark for evaluating multimodal models. It adopts a three-level ability taxonomy. The first level, L-1, measures two high-level capabilities, perception and reasoning. The second level, L-2, refines these into six sub-abilities. The third level, L-3, provides the most detailed view by specifying 20 fine-grained ability dimensions, enabling a comprehensive analysis of model behavior.

**MME** (Fu et al., 2023). MME serves as a broad benchmark for measuring multimodal model capabilities from multiple perspectives. It consists of 14 subtasks that separately test different aspects of perception and reasoning. The evaluation is formulated with instruction and answer pairs, using concise instructions to reduce potential leakage and improve consistency, which supports a more reliable and fair comparison across models.

**POPE** (Li et al., 2023). POPE is designed to quantify object hallucination in vision language models. It evaluates whether a model incorrectly claims that an object exists in an image by asking targeted yes or no questions about object presence. Performance is summarized with Accuracy, Recall, Precision, and F1 under three sampling protocols, which together offer a reliable view of hallucination tendencies and object grounded behavior.

**ScienceQA** (Lu et al., 2022). ScienceQA is a broad science question answering benchmark covering disciplines such as natural science, language-related subjects, and social science. It organizes questions with a hierarchical label system that includes topics, categories, and skills, totaling 26 topics, 127 categories, and 379 skills. This structure supports diverse problem types and enables evaluation of multimodal comprehension, multi-step reasoning, and interpretability.

**VQA-V2** (Goyal et al., 2017). VQA-v2 is a large-scale visual question answering benchmark that tests visual understanding using open-ended questions about natural images. It contains 265,016 images covering diverse everyday scenes and objects. For each question, the dataset provides 10 human-annotated reference answers, which supports robust scoring and reduces sensitivity to individual annotator variation.

**TextVQA** (Singh et al., 2019). TextVQA targets settings where answering a question requires reading text that appears inside the image. The task evaluates whether a model can combine visual understanding with text recognition and use the extracted words to support reasoning. Questions therefore depend on both scene content and embedded textual cues, rather than purely visual signals.

**MSVD-QA** (Xu et al., 2017). MSVD-QA is a video question answering benchmark built from the Microsoft Research Video Description (MSVD) dataset. It contains 1,970 short video clips and about 50.5K associated question-answer pairs. The questions cover diverse aspects of the video content and are commonly used to evaluate video QA, and in some settings also video captioning related capabilities. Question types are grouped into five classes, namely what, who, how, when, and where.

**MSRVTT-QA** (Xu et al., 2017). MSRVTT-QA is a large-scale video QA benchmark comprising 10K videos and roughly 243K question-answer pairs. The dataset emphasizes reasoning over dynamic content, so accurate answers often depend on capturing both appearance information and temporal evolution across frames. As in MSVD-QA, questions are organized into five types, including what, who, how, when, and where, which supports fine-grained evaluation by question category.

### B.2. Implementation Details

All of our experiments are conducted on Nvidia A6000 GPU. The implementation was carried out in Python 3.10, utilizing PyTorch 2.1.2, and CUDA 12.1. All baseline settings follow the original paper. For our method, we set the number of diffusion steps for energy propagation to 2. In the quadtree partitioning process, we configure the minimum height and width of each crop (denoted as $m$ in Eq. 8) to 4, which defines the finest granularity of spatial division.

## C. Detailed Computational Complexity Analysis

In this appendix, we provide a detailed empirical analysis of the computational cost of SpecFlow, complementing the formal treatment in Sec. 4.4. We report (i) end-to-end wall-clock latency, prefill time, and GPU memory, (ii) a fine-grained breakdown of the pruning-stage overhead, and (iii) a FLOPs comparison under aggressive pruning.

### C.1. End-to-End Wall-Clock Profiling

We measure practical efficiency on the NVIDIA RTX A6000 GPU using LLaVA-1.5-7B. We report prefill time, end-to-end latency, and peak GPU memory. Two pruning ratios are considered: a moderate setting (192 retained tokens, 66.7% pruning)

and an aggressive setting (64 retained tokens, $88.9\%$ pruning). Results are summarized in Table 5.

*Table 5.* End-to-end efficiency of SpecFlow versus baselines on LLaVA-1.5-7B.

| Method | Tokens | Prune % | Prefill (ms) | Latency (s) | Mem. (GB) |
|---|---|---|---|---|---|
| Vanilla | 576 | 0.0 | 86.6 | 0.397 | 19.2 |
| FastV | 192 | 66.7 | 42.0 | 0.284 | 16.2 |
| HoloV | 192 | 66.7 | 22.0 | 0.248 | 15.8 |
| **SpecFlow (Ours)** | 192 | 66.7 | 24.5 | 0.259 | 15.9 |
| FastV | 64 | 88.9 | 29.0 | 0.249 | 15.8 |
| HoloV | 64 | 88.9 | 14.0 | 0.221 | 14.8 |
| **SpecFlow (Ours)** | 64 | 88.9 | 15.5 | 0.229 | 14.8 |

Two observations follow. First, SpecFlow achieves substantially faster inference than the unpruned baseline: at $88.9\%$ pruning, end-to-end latency drops from $0.397\,\mathrm{s}$ to $0.229\,\mathrm{s}$ (a $1.73\times$ speedup) and GPU memory drops from $19.2\,\mathrm{GB}$ to $14.8\,\mathrm{GB}$. Second, under the same token budget, SpecFlow introduces only a modest additional overhead relative to the strongest pruning baseline HoloV ($+2.5\,\mathrm{ms}$ prefill at 192 tokens and $+1.5\,\mathrm{ms}$ at 64 tokens), while consistently delivering higher accuracy than HoloV under our reported settings.

## C.2. Breakdown of the Pruning-Stage Overhead

To attribute the runtime cost of SpecFlow to its individual components, we further measure the wall-clock time of each stage: $k$NN graph construction, spectral heat diffusion, and the combined quadtree partitioning, Top-$K$ selection, and sink-token construction. Results are reported in Table 6.

*Table 6.* Wall-clock breakdown of the SpecFlow pruning module on RTX A6000 with LLaVA-1.5-7B. The total pruning overhead is small relative to the end-to-end latency.

| Setting | $k$NN Graph | Diffusion | Quadtree+TopK+Sink | Total | Latency |
|---|---|---|---|---|---|
| 192 tokens | 1.1 ms | 0.6 ms | 0.9 ms | 2.6 ms | 0.259 s |
| 64 tokens | 1.1 ms | 0.6 ms | 1.2 ms | 2.9 ms | 0.229 s |

In both settings, the total pruning overhead is at most $2.9\,\mathrm{ms}$, which corresponds to roughly $1\%$ of the end-to-end latency ($0.229$–$0.259\,\mathrm{s}$). The $k$NN graph and diffusion costs are essentially constant across pruning ratios, while the quadtree/Top-$K$/sink stage scales mildly with the chosen budget. This confirms that the pruning module is not the runtime bottleneck: the dominant cost still resides in the downstream LLM execution, particularly the decoding stage.

## C.3. FLOPs Comparison Under Aggressive Pruning

To complement wall-clock measurements with a hardware-agnostic metric, we report total inference FLOPs at the 64-token setting in Table 7. The unpruned LLaVA-1.5-7B requires $8.12\,\mathrm{T}$ FLOPs, while SpecFlow reduces this to $1.26\,\mathrm{T}$ FLOPs ($-84.5\%$). SpecFlow also consumes fewer FLOPs than FastV at the same token budget ($1.26\,\mathrm{T}$ vs. $1.64\,\mathrm{T}$), even after accounting for the additional graph-diffusion and sink-construction operations introduced by our method.

*Table 7.* Total inference FLOPs at the 64-token setting on LLaVA-1.5-7B.

| Method | FLOPs |
|---|---|
| LLaVA-1.5-7B (vanilla) | 8.12 T |
| FastV (64 tokens) | 1.64 T |
| **SpecFlow (64 tokens)** | **1.26 T** |

## C.4. Discussion

Combining the three measurements above, we draw the following conclusions:

- **Substantial inference speedup.** At $88.9\%$ pruning, SpecFlow reduces end-to-end latency from $0.397\,\mathrm{s}$ to $0.229\,\mathrm{s}$ and

peak GPU memory from $19.2$ GB to $14.8$ GB, while retaining $95.6\%$ of the unpruned model's accuracy.

- **Negligible pruning overhead.** The combined cost of $k$NN graph construction, spectral diffusion, and quadtree/sink-token construction stays within $2.6$–$2.9$ ms, i.e., about $1\%$ of the total end-to-end latency. The added structural modules in SpecFlow are therefore not the runtime bottleneck.

- **Favorable FLOPs.** Despite introducing graph-based propagation and sink summarization, SpecFlow requires fewer total FLOPs than FastV at the same 64-token budget ($1.26$ T vs. $1.64$ T), and only $15.5\%$ of the FLOPs of the unpruned model.

Overall, these measurements indicate that SpecFlow is competitive with the strongest pruning baseline (HoloV) in runtime while achieving better accuracy, and it remains efficient in terms of latency, memory, and FLOPs. The added structural mechanisms do not become a practical inference bottleneck.

## D. Visualization of Attention and Energy Distribution

To further motivate our design of energy diffusion (HeatFlow), we visualize the distribution and spatial consistency of both raw [CLS] attention scores and our diffused energy scores. This visualization directly addresses the limitations of attention-based pruning highlighted earlier. As shown in Figure 6 (left), raw [CLS] attention yields a spiky distribution that concentrates on a small set of high-value tokens and leaves most tokens with negligible weights. This spiky pattern aligns with the brittle point-wise ranking of existing methods, which risks spatial fragmentation and contextual loss during token pruning. In contrast, Figure 6 (right) shows that HeatFlow transforms this distribution. The density curve spreads across a broader range of values (zoom in for better visualization), and the spatial overlay extends high-energy signals to coherent semantic regions instead of isolated points. This smoothing effect encodes the spatial coherence and contextual dependency of visual tokens, which are ignored by naive attention-based pruning. It ensures our importance scores support robust and structure-preserving token selection.

## E. Limitations and Future Work

While SpecFlow achieves strong accuracy and efficiency trade-offs across multiple benchmarks and token budgets, several limitations remain. First, our current evaluation emphasizes accuracy under fixed token budgets and provides FLOPs-based analysis, end-to-end wall-clock measurements (latency, throughput, and peak memory) can depend on hardware, batch size, FlashAttention settings, and the overhead of preprocessing (kNN graph construction, diffusion, and quadtree selection). A thorough systems study that profiles these costs across resolutions and long-video regimes is an important next step. Second, SpecFlow relies on access to intermediate representations (visual token embeddings and attention from the vision encoder), this requirement may limit applicability to black-box VLM APIs or architectures that do not expose reliable CLS attention. Future work could explore more robust or model-agnostic seeding signals (e.g., multi-layer/ multi-head aggregation rules, text-guided cues, or lightweight learned selectors) and adaptive hyperparameters (e.g., diffusion strength/steps, kNN sparsity, and quadtree thresholds) that generalize across backbones without tuning.

Third, our coverage-aware pruning currently assumes a 2D token grid and uses a spatial quadtree. For video inputs, temporal structure is not explicitly modeled, and extending the method to spatiotemporal partitioning or 3D diffusion could further improve stability on dynamic scenes. Finally, the proposed mean + residual sink tokens provide a very compact summary of pruned content, which may be insufficient for inputs requiring fine-grained evidence (e.g., small objects, dense text/OCR, counting, or rare attributes). Exploring adaptive numbers of sink tokens, stronger coreset constructions, or task-aware condensation could better preserve rare but critical cues under extreme compression. We also plan to standardize token-budget accounting (including any appended sinks) and expand analyses of failure cases to clarify when conservative condensation may still break down.

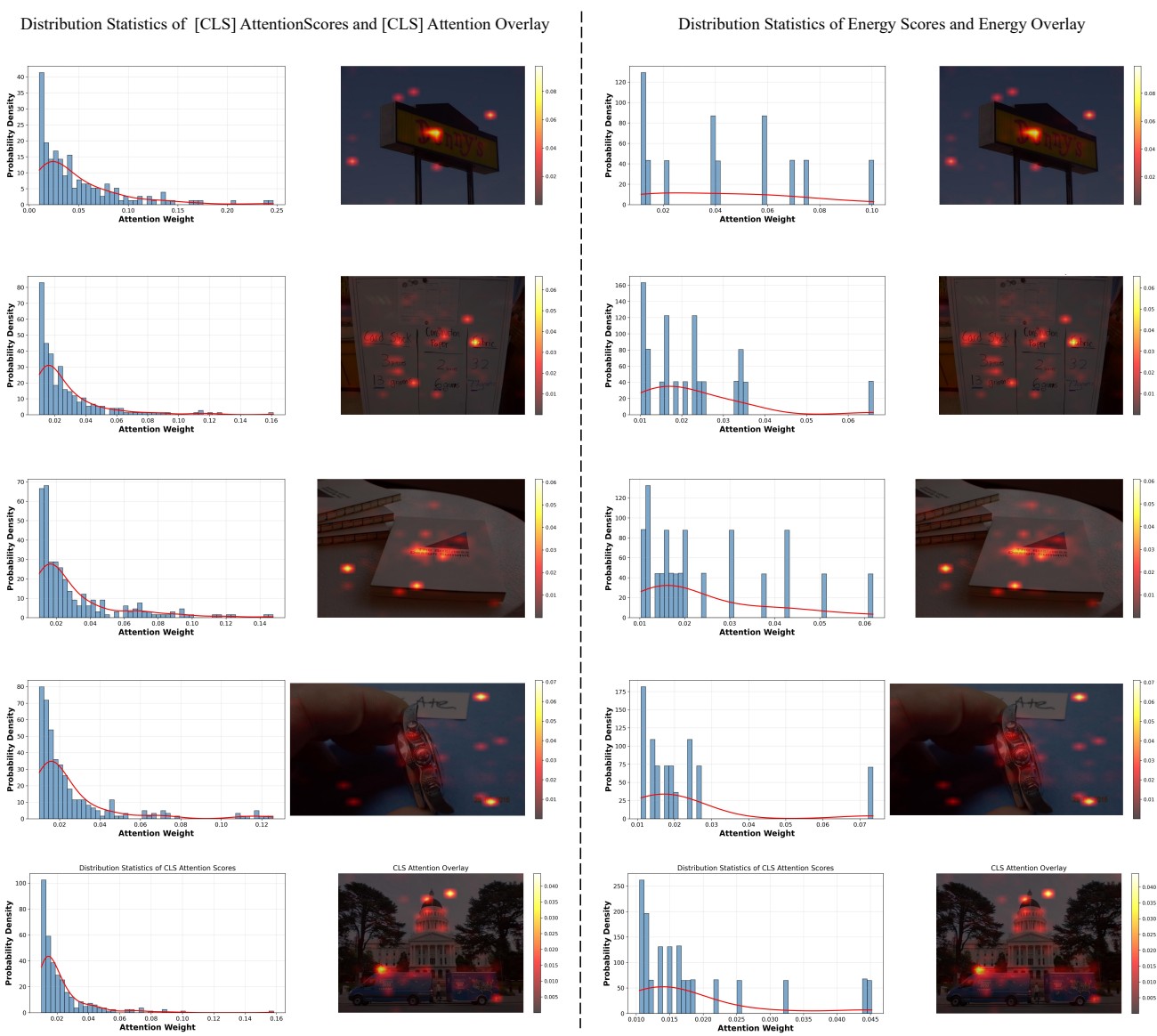

*Figure 6.* Visualization of raw [CLS] attention and HeatFlow-diffused energy. Left: Raw attention exhibits a spiky distribution. Right: Diffused energy is smoothed and covers coherent semantic regions.

