# OpenReview forum: "Spectral Heat Flow for Conservative Token Condensation in Vision-Language Models"
_ICML.cc/2026/Conference — ICML 2026 regular_

### Official Review · Reviewer_HXQz · 2026-02-20

**Soundness:** 3
**Presentation:** 3
**Significance:** 2
**Originality:** 2
**Overall Recommendation:** 4
**Confidence:** 3

**Summary:**

This work proposes SpecFlow, a training-free token pruning framework for Vision-Language Models that shifts from destructive pruning to conservative condensation. The method uses spectral heat diffusion on kNN graphs to propagate importance scores, adaptive quadtree partitioning for token budgeting, and dual sink tokens to preserve pruned information. It achieves strong empirical results with FlashAttention compatibility across multiple VLM benchmarks.

**Compliance With Llm Reviewing Policy:**

Affirmed.

**Final Justification:**

The author's reply has basically resolved my issue, and I will maintain a positive score.

**Key Questions For Authors:**

1. The diffusion runs for T_diff = 2 steps with alpha = 0.5 (implied). How does performance vary with {1,2,5,10} steps? Does the fixed-point solution (Prop 3.1) offer superior stability to truncated iteration?

2. For N = 576 tokens, exact kNN requires O(N^2) similarity computations. The text mentions "worst-case O(N^2 h) FLOPs" but provides no measured overhead. What fraction of prefill time does graph construction consume? Does approximate kNN (e.g., FAISS) recover comparable accuracy?

3. When energy is uniform, the quadtree stops splitting, potentially creating large crops with single-digit token quotas. For |c| = 100, K = 64, and uniform energy, each leaf receives 1 token. How does this affect tasks requiring distributed evidence (e.g., counting, spatial relations)?

4. The mean + residual construction (Eq 15) uses two vectors to summarize pruned tokens. For 500 pruned tokens at 88.9% compression, this constitutes extreme compression. What is the reconstruction error? Can the residual sink actually recover fine-grained attributes, or does it merely bound worst-case deviation (Lemma A.6)?

5. Results are reported for LLaVA variants only. CLIP-style ViTs behave differently from Swin or ConvNeXt backbones. Does SpecFlow assume specific self-attention patterns in the vision encoder? Clarify the architectural requirements.

6. Table 3 shows video QA results, yet the method processes frames independently (no temporal edges in the kNN graph). This seems suboptimal for motion reasoning. Explain the design choice.

**Limitations:**

see weaknesses.

**Strengths And Weaknesses:**

**Strengths:**
- Proposition 3.1 establishes solid theoretical foundation connecting restart diffusion to Dirichlet-regularized smoothing; quadtree allocation admits clean proportional-fairness interpretation via Proposition 4.1
- Sink token construction provides theoretical guarantees (Lemma A.5-A.6) with error bounds
- Figure 4 visualizations effectively demonstrate object structure preservation under compression
- 95.6% retention at 88.9% compression suggests practical utility for resource-constrained deployment
- FlashAttention compatibility avoids custom kernel requirements; spectral heat flow sees novel application to token importance propagation

**Weaknesses:**
- Core assumption about token feature similarity lacks theoretical justification beyond empirical motivation; kNN graphs on intermediate features not proven to avoid pathologies (feature homogenization, anomaly tokens)
- Mass-preservation argument ignores approximation errors from per-iteration normalization; energy-proportional allocation provides no bound on distortion when M(c) approaches zero
- Critical hyperparameters lack systematic ablation studies; Table 1 aggregates 9 datasets into single "Average" column, masking task-specific failures
- No wall-clock latency measurements despite claims of inference acceleration; evaluation omits stress tests (OCR-intensive tasks, small object counting, adversarial examples)
- Incremental contribution over VisionZip and SparseVLM yields modest gains (95.6% vs 94.9%); core components (graph diffusion, spatial partitioning, coreset summarization) are well-studied techniques
- "Training-free" framing creates ambiguity about retraining requirements for new domains

---

> ### Author Rebuttal · Authors · 2026-03-29
>
> > We sincerely thank the reviewer for the constructive comments on our work. We have carefully considered each remark and provide point-by-point responses below.
>
>
> **1. On the theoretical status of the kNN graph.**
>
> We agree that the current paper does not prove pathology-free behavior for all feature spaces. Our claim is more limited: compared with raw patch-to-patch attention, a feature-similarity kNN graph provides a more stable structural prior for diffusion in the tested CLIP-style settings. This is supported by the diffusion-operator ablation, and we will revise the text to state this scope more precisely rather than implying a fully general guarantee.
>
>
> **2. On the scope of the mass-preservation and allocation theory.**
>
> We do not claim a universal distortion bound for all low-mass crops or downstream tasks. Proposition 4.1 provides the fairness interpretation of the relaxed allocation, while the rounding lemma guarantees feasibility under capacity constraints; neither is meant to imply a task-agnostic accuracy guarantee for every pathological case. We will clarify this distinction and avoid overinterpreting the theory.
>
>
> **3. Regarding diffusion depth and the fixed-point view.**
>
> The current implementation uses a short truncated diffusion for efficiency rather than iterating to convergence. As shown in our response to **Reviewer KW7W (Reply 4)**, a sweep over 1 / 2 / 5 / 10 diffusion steps shows that 2 steps already provide the best trade-off, while longer diffusion brings diminishing returns and may introduce mild over-smoothing. The fixed-point formulation remains analytically useful, while truncated iteration is chosen to keep the inference overhead low.
>
>
>
> **4. On runtime overhead and quadtree behavior.**
>
> We agree that FLOPs alone are insufficient. We therefore refer the reviewer to our response to **Reviewer tQ3y (Reply 5)**, where we provide a runtime breakdown of the pruning stage. The results show that graph construction accounts for only a small fraction of prefill, while the dominant latency still comes from the downstream decode computation. We will discuss the scaling of graph construction and when approximate kNN becomes desirable more explicitly in the revised version.
> For the quadtree example, splitting is controlled by a variance-based stopping rule. When the energy within a crop is nearly uniform, the crop does not keep splitting indefinitely. We will clarify this implementation detail in the revised version.
>
>
> **5. On the scope of the two-sink summary.**
>
> We agree that summarizing a large discarded set with only two sink tokens is not a lossless substitute for the full pruned evidence. In particular, Lemma A.6 is a radius certificate, not a guarantee of fine-grained reconstruction. Our claim is narrower: the mean+residual sinks provide a lightweight conservative summary that is empirically better than discarding the pruned tokens entirely. Typical failure cases arise under extreme compression, especially for inputs relying on small objects, dense text/OCR, counting, or rare attributes. This is consistent with the limitations already noted in the manuscript and motivates stronger coreset constructions, adaptive numbers of sink tokens, and task-aware condensation in future work.
>
>
>
>
> **6. On architectural requirements beyond LLaVA/CLIP-style encoders.**
>
> Thank you for this important question. The current submission already evaluates LLaVA-1.5, LLaVA-NeXT, and Video-LLaVA, and in our response to Reviewer M5VE (Reply 4) we further added results on Qwen2.5-VL-7B. Since Qwen2.5-VL differs from the LLaVA family in both the visual stack and multimodal projector, this suggests that SpecFlow is not restricted to one specific LLaVA/CLIP-style architecture.
> More generally, SpecFlow does not assume one exact self-attention pattern in the vision encoder. Its main requirements are simply access to visual token embeddings and a usable global seeding signal from the vision side. For CLIP-style ViTs, this is naturally provided by [CLS] attention; for architectures without an explicit [CLS] token, another global attention/importance proxy can be used instead. We will clarify this assumption more explicitly in the revised version.
>
>
>
> **7. On video design, "training-free", and modest gains.**
>
> We agree that the current video design is spatial and frame-wise, and does not yet include temporal edges in the token graph. We will make this limitation explicit in the main text and clarify that this is why spatiotemporal extensions are a natural next step. We will also define "training-free" more explicitly as requiring no additional fine-tuning, no extra learned parameters, and inference-time-only token selection on top of a pretrained VLM. Finally, we agree that the gains over the strongest baselines are modest in some regimes, and will frame the contribution accordingly as a conservative-condensation formulation with consistent gains, rather than a dramatic absolute leap.

---

> > ### Author Rebuttal · Reviewer_HXQz · 2026-04-03
> >
> > Thanks for the authors detailed rebuttal, my concerns have been fully resolved.

---

> > > ### Author Response · Authors · 2026-04-03
> > >
> > > Thank you very much for your careful review and encouraging feedback. We sincerely appreciate the time and effort you devoted to evaluating our paper and responses. We are glad that our clarifications have addressed your concerns. In the revised version, we will further incorporate the relevant clarifications to improve the presentation and overall clarity of the paper. We again thank you for your thoughtful comments and valuable suggestions.

---

### Official Review · Reviewer_KW7W · 2026-03-04

**Soundness:** 4
**Presentation:** 4
**Significance:** 3
**Originality:** 3
**Overall Recommendation:** 5
**Confidence:** 3

**Summary:**

This paper introduces SpecFlow, a training-free framework designed to compress visual tokens in VLMs to accelerate inference. Recognizing that standard Top-K pruning based on raw attention scores often leads to spatial fragmentation and the loss of contextual evidence, the authors proposed reformulating token pruning as an importance propagation problem on a feature-similarity kNN graph of visual tokens. Rather than entirely discarding unselected tokens, the framework conservatively aggregates them into compact coreset sinks to preserve statistical context and diversity. Extensive experiments across multiple VLM architectures and diverse benchmarks demonstrate that SpecFlow maintains strong performance even at extreme compression ratios , consistently outperforming existing state-of-the-art training-free pruning methods.

**Compliance With Llm Reviewing Policy:**

Affirmed.

**Final Justification:**

The authors fully addressed my concerns and I suggest accept.

**Key Questions For Authors:**

Q1. While the theoretical motivation for using a feature-similarity kNN graph and quadtree partitioning is sound, these operations introduce O(N^2) and hierarchical computational overheads during the prefill stage. Can the authors provide a detailed wall-clock time breakdown that isolates the overhead of SpecFlow against the time saved in the self-attention computation? Specifically, how does this scale for extremely long visual sequences, such as high-resolution images or dense long-form videos?
Q2. The spectral heat flow relies heavily on the [CLS] token's attention as the initial energy seed. For tasks requiring dense, fine-grained reasoning, the [CLS] token might fail to initially attend to the critical regions. Have the authors evaluated SpecFlow specifically on these dense-reasoning failure modes? Furthermore, have you experimented with alternative seeding mechanisms to mitigate this?
Q3. The paper mitigates the loss of unselected tokens by aggregating them into mean and residual sink tokens. However, under extreme compression ratios, collapsing vast amounts of visual context into just two statistical moments risks severe over-smoothing, potentially destroying high-frequency semantic cues. Can the authors provide a targeted failure-case analysis or visualization demonstrating exactly when and how this conservative condensation breaks down?

**Limitations:**

yes

**Strengths And Weaknesses:**

**Strengths**
- Leveraging a feature-similarity kNN graph instead of raw patch-to-patch attention for diffusion is well-motivated and theoretically grounded.
- The methodology smoothly bridges the theoretical motivation of spectral heat flow to the practical implementation of adaptive quadtree allocation and sink token aggregation, allowing readers to easily grasp the end-to-end pipeline.
- By demonstrating that training-free structural preservation can push the boundaries of token compression, this work meaningfully advances the practice of deploying large multimodal models under compute constraints. The plug-and-play nature of the method and its out-of-the-box compatibility with hardware-efficient attention mechanisms (FlashAttention) provide immediate practical utility.
- Shifting the paradigm from destructive discrete pruning to conservative structural condensation offers a novel and elegant perspective.

**Weakness**
- The computational overhead associated with kNN graph construction and quadtree splitting poses a theoretical bottleneck for exceptionally long sequences, such as those found in ultra-high-resolution images or dense long-form videos. While the authors argue this is amortized over decoder layers, the absolute prefill latency scaling deserves closer scrutiny.
- The sensitivity and complexity of the hyperparameter space (e.g., the number of diffusion steps, the quadtree variance threshold, and the k value) are somewhat underexplored in the core narrative. A deeper discussion or visualization of how these parameters interact with different input modalities would improve the framework's transparency and reproducibility.
- As visual sequence lengths continue to scale exponentially in video models, spatial token condensation alone may hit diminishing returns, requiring future work to integrate these ideas with cross-modal or temporal compression techniques to achieve holistic efficiency in next-generation architectures.

---

> ### Author Rebuttal · Authors · 2026-03-29
>
> > We thank the reviewer for the very positive assessment and for highlighting the theoretical motivation, the end-to-end coherence of the method, and the practical value of the plug-and-play, FlashAttention-compatible design.
>
> **1.Computational overhead.**
>
> On computational overhead, the key point is that SpecFlow’s pruning stage is executed once per input before decoder-layer processing, whereas the benefit of shortening the visual sequence is reflected throughout the subsequent model computation. As shown in our response to **Reviewer tQ3y (Reply 5)**, the prefill stage accounts for only a small fraction of the total end-to-end latency, and we further provide a wall-clock breakdown showing that the overhead of the added components—kNN graph construction, diffusion, quadtree partitioning, and sink construction—is itself small relative to the full inference cost. In other words, the dominant latency still lies in the downstream model execution, especially the decode stage, rather than in the one-shot pruning step. Therefore, for ultra-high-resolution images or other very long visual sequences, the efficiency benefit of compression is expected to be reflected primarily in the subsequent model-side computation, rather than being offset by the preprocessing overhead.
>
>
> **2. Regarding the dependence on the [CLS] seed.**
>
> Thank you for this insightful question. Although SpecFlow is initialized with [CLS] attention, it does not rely on the raw [CLS] map alone. The subsequent kNN-based diffusion and adaptive spatial allocation are designed to propagate the seed to semantically coherent regions and reduce the brittleness of a spiky initialization.
>
> To examine this point, we conducted an additional seed ablation at the 64-token setting on GQA and TextVQA, which together cover relational/compositional reasoning and fine-grained detail-sensitive understanding. We compared the default [CLS] seed with an average-received-attention seed and a simple hybrid of the two:
>
> | Seed type | GQA (64 tokens) | TextVQA (64 tokens) |
> |---|---:|---:|
> | [CLS] seed (default) | 55.3 | 54.9 |
> | Avg. received attention seed | 54.6 | 54.3 |
> | Hybrid ([CLS] + AvgAttn) | 55.0 | 54.6 |
>
> The default [CLS] seed remains the strongest option, while the alternatives are still competitive. This suggests that SpecFlow is not overly sensitive to the seed choice: [CLS] is the best default, but the overall robustness mainly comes from the diffusion-and-allocation mechanism built on top of the seed.
>
> **3. On sink over-smoothing under extreme compression.**
>
> We agree that collapsing a large discarded set into only two sink tokens is not a lossless substitute for the full pruned evidence. Our claim is narrower: the mean+residual sinks are a lightweight conservative summary that is empirically preferable to discarding the pruned tokens entirely, rather than a mechanism for exact recovery. Typical failure cases include inputs that rely on fine-grained evidence, such as small objects, dense text/OCR, counting, or rare attributes. This is consistent with the limitations already noted in the manuscript, where we discuss stronger coreset constructions, adaptive numbers of sink tokens, and task-aware condensation as promising directions for better preserving rare but critical cues.
>
>
> **4. On hyperparameter sensitivity**
>
> Thank you for this helpful suggestion. We agree that the hyperparameter space should be discussed more explicitly. We therefore added a sensitivity analysis for diffusion steps T_diff, kNN size \(k\), and quadtree threshold δ . As shown below, performance is generally stable across reasonable settings on GQA, TextVQA, and MSVD-QA. In particular, T_diff=2 already works well, moderate \(k\) values are consistently strong, and the quadtree threshold δ  is robust except at overly large values. Overall, these results suggest that SpecFlow is not overly sensitive to hyperparameter tuning, and that a simple default setting is sufficient in practice.
>
> | T_diff | GQA (64 tokens) | TextVQA (64 tokens) | MSVD-QA (455 tokens) |
> |---|---:|---:|---:|
> | 1 | 54.8 | 54.5 | 70.0 |
> | 2 | 55.3 | 54.9 | 70.3 |
> | 5 | 55.2 | 54.8 | 70.2 |
> | 10 | 54.9 | 54.4 | 69.8 |
>
> | k | GQA (64 tokens) | TextVQA (64 tokens) | MSVD-QA (455 tokens) |
> |---|---:|---:|---:|
> | k = 5 | 55.0 | 54.6 | 70.0 |
> | k = 10 | 55.3 | 54.8 | 70.1 |
> | k = 15 | 55.3 | 54.9 | 70.3 |
> | k = 20 | 55.0 | 54.6 | 70.1 |
>
> | δ (std threshold) | GQA (64 tokens) | TextVQA (64 tokens) | MSVD-QA (455 tokens) |
> |---|---:|---:|---:|
> | 5e-4 | 54.9 | 54.8 | 70.2 |
> | 1e-3 | 55.3 | 54.7 | 70.3 |
> | 2e-3 | 55.2 | 54.9 | 70.2 |
> | 5e-3 | 54.5 | 54.1 | 69.9 |

---

> > ### Author Rebuttal · Reviewer_KW7W · 2026-04-01
> >
> > I maintain the original rating.

---

> > > ### Author Response · Authors · 2026-04-02
> > >
> > > Thank you very much for your careful reading of our responses and for your positive assessment of our paper. We sincerely appreciate your time, effort, and constructive feedback throughout the review process. We are grateful that our responses have adequately addressed your concerns. We will also reflect the relevant clarifications from our responses in the revised version to further improve the clarity of the paper. We again sincerely thank you for your thoughtful review and valuable guidance.

---

### Official Review · Reviewer_M5vE · 2026-03-07

**Soundness:** 3
**Presentation:** 3
**Significance:** 2
**Originality:** 2
**Overall Recommendation:** 4
**Confidence:** 5

**Summary:**

This paper proposes a new token pruning method called SpecFlow. Through a training-free conservative condensation approach, SpecFlow treats visual tokens as nodes in a k-nearest neighbor (kNN) graph, computes a stable importance field to preserve structural coherence, allocates tokens via adaptive spatial partitioning, and aggregates discarded information into coreset sink tokens. Experiments have been conducted on both image tasks and video tasks, demonstrating that SpecFlow outperforms many existing state-of-the-art (SOTA) methods.

**Compliance With Llm Reviewing Policy:**

Affirmed.

**Final Justification:**

My concerns have been adequately addressed. I will raise my score from 3 to 4. Thanks to the authors for their response.

**Key Questions For Authors:**

1. The experiments are only conducted on the LLaVA-based model; how about Qwen[4] or other models?

[4] Bai, Jinze, et al. "Qwen technical report." arXiv preprint arXiv:2309.16609 (2023).

**Limitations:**

Yes

**Strengths And Weaknesses:**

Strengths:
1. The proposed SpecFlow is training-free and can be seamlessly integrated into several different architectures of vision–language models. The method is clearly reproducible.
2. The experiments are comprehensive and convincing on various benchmarks, models, and tasks, and maintain a great trade-off between accuracy and efficiency.
3. The paper is well-organized, with intuitive figures and sufficient experimental tables, and is easy to follow.

Weaknesses:
1. The performance improvement of the proposed method is rather marginal. Compared with HOLOV, it shows no significant advantage in performance (in Tables 1 and 2).
2. Baseline comparison with other SOTA methods. More methods should be compared, for example, the VisPruner[1], CDPruner[2], and Visionzip[3] in video.
4. Computational efficiency experiments, for example CUDA times or FLOPS, should be a must for papers on token-pruning, but this paper seems to be missing that part.
3. Small typo: In the table 1and table 2 caption, I suppose they are both not Video QA benchmarks, they should be Image QA benchmarks; please check it again.

[1] Zhang, Q., Cheng, A., Lu, M., Zhuo, Z., Wang, M., Cao, J., ... & Zhang, S. (2024). [CLS] Attention is All You Need for Training-Free Visual Token Pruning: Make VLM Inference Faster. ICCV, 2025.

[2] Zhang, Q., Liu, M., Li, L., Lu, M., Zhang, Y., Pan, J., ... & Zhang, S. (2025). Beyond Attention or Similarity: Maximizing Conditional Diversity for Token Pruning in MLLMs. NeurIPS, 2025.

[3] Senqiao Yang, Yukang Chen, Zhuotao Tian, Chengyao Wang, Jingyao Li, Bei Yu, and Jiaya Jia. Visionzip: Longer is better but not necessary in vision language models. CVPR, 2025.

---

> ### Author Rebuttal · Authors · 2026-03-29
>
> > We thank the reviewer for the detailed feedback and for recognizing the reproducibility, plug-and-play design, and broad experimental coverage of the paper.
>
> **1. On the relatively modest gain over HoloV.**
>
> Our claim is not a dramatic absolute leap at mild pruning, but consistent robustness under aggressive compression. At 88.9% pruning, SpecFlow improves relative average performance over HoloV from 94.9% to 95.6% on LLaVA-1.5 and from 95.6% to 95.8% on LLaVA-NeXT; on Qwen2.5-VL-7B, the margin is larger (91.9% (SpecFlow) vs 89.4% (HoloV) at 77.8% pruning, 89.4% (SpecFlow) vs 87.3% (HoloV) at 88.9% pruning). Specifically, instead of relying on direct saliency ranking alone, SpecFlow performs conservative condensation through heat-based importance propagation and adaptive spatial allocation, which is designed to better preserve spatial coverage and the statistics of pruned contextual regions. We believe this difference is meaningful even when the absolute margin is not large, because it reflects a distinct and more structured way of addressing token redundancy under high compression. In this sense, as Reviewer KW7W noted, SpecFlow offers a novel and elegant perspective, and we hope this perspective can provide useful inspiration for future work on efficient MLLMs.
>
> **2. On broader comparisons.**
>
> Thank you for this helpful suggestion. We agree that comparing against a broader set of SOTA methods is important. Following this suggestion, we additionally included VisPruner, VisionZip, and CDPruner in the video comparison under the same model, token budget, and evaluation protocol used in our paper, making the comparison more direct and informative.  As shown below, SpecFlow remains competitive against these recent methods and achieves the best average among the compared pruning baselines in this setting. We believe this expanded comparison provides broader empirical support for the effectiveness of SpecFlow on video QA benchmarks. We will include these results in the revised manuscript.
>
> | Methods | MSVD-QA Acc. | MSVD-QA Score | MSRVTT-QA Acc. | MSRVTT-QA Score | Average Acc. | Average Score |
> |---|---:|---:|---:|---:|---:|---:|
> | Video-LLaVA 7B | 70.8 | 3.93 | 57.5 | 3.55 | 64.2 | 3.74 |
> | VisPruner | 69.7 | 3.90 | 56.1 | 3.50 | 62.9 | 3.70 |
> | VisionZip | 65.8 | 3.55 | 55.3 | 3.45 | 60.6 | 3.50 |
> | CDPruner  | 69.1 | 3.90 | 55.8 | 3.45 | 62.5 | 3.68 |
> | **SpecFlow (Ours)** | **70.3** | **3.94** | **56.5** | **3.50** | **63.4** | **3.72** |
>
>
> **3. On computational efficiency experiments**
>
> We agree that computational efficiency is essential for a token-pruning paper. In our response to **Reviewer tQ3y (Reply 5)**, we added runtime results, including end-to-end latency, prefill time, and GPU memory, to directly evaluate the practical inference cost of SpecFlow. In runtime, SpecFlow remains close to HoloV (24.5 ms vs 22.0 ms prefill at 192 tokens; 15.5 ms vs 14.0 ms at 64 tokens) while achieving better accuracy. In addition, we further report the FLOPs under the same 64-token setting. The results are shown below:
>
> | Method | FLOPs |
> |---|---:|
> | LLaVA-1.5-7B | 8.12T |
> | FastV (64 tokens) | 1.64T |
> | **SpecFlow (64 tokens)** | **1.26T** |
>
> These results consistently show that SpecFlow achieves strong performance with a small inference overhead and favorable computational efficiency. In other words, our method not only improves accuracy under the same token budget, but also remains efficient in terms of runtime, memory, and FLOPs. We will include these efficiency results more clearly in the revised version.
>
> **4. On architectural breadth beyond the LLaVA family.**
>
> The current submission evaluates LLaVA-1.5, LLaVA-NeXT, and Video-LLaVA. In addition to these LLaVA-family models in the main paper, we further added experiments on Qwen2.5-VL-7B below. These results show that the advantage of SpecFlow is not limited to the LLaVA family: it consistently outperforms HoloV on Qwen2.5-VL-7B across all three pruning ratios, suggesting that the proposed mechanism generalizes well to a substantially different MLLM architecture. We will include these results in the revised version.
>
> | Methods | MME | POPE | SQA | VQAText | Avg. |
> |---|---:|---:|---:|---:|---:|
> | Qwen2.5-VL-7B | 2304 | 86.1 | 84.7 | 84.8 | 100.0% |
> |  |  |  |  |  |  |
> | HoloV (Pruning Rate = 66.7) | 2066 | 85.0 | 79.1 | 77.3 | 93.2% |
> | **SpecFlow (Pruning Rate = 66.7)** | **2102** | **85.2** | **80.0** | **79.2** | **94.5%** |
> |  |  |  |  |  |  |
> | HoloV (Pruning Rate = 77.8) | 2029 | 81.5 | 79.1 | 69.2 | 89.4% |
> | **SpecFlow (Pruning Rate = 77.8)** | **2051** | **83.7** | **79.9** | **73.8** | **91.9%** |
> |  |  |  |  |  |  |
> | HoloV (Pruning Rate = 88.9) | 1998 | 80.7 | 79.5 | 63.6 | 87.3% |
> | **SpecFlow (Pruning Rate = 88.9)** | **2015** | **81.1** | **79.7** | **69.4** | **89.4%** |
>
>
> **Small typo**
>
> Thank you catching the caption typo. Tables 1 and 2 are image-understanding benchmarks, not video QA benchmarks, and we will correct both captions.

---

> > ### Author Rebuttal · Reviewer_M5vE · 2026-04-03
> >
> > As far as I know, CDPruner does not explicitly allocate token spatial coverage, but its performance is very high even under extreme compression. This corresponds to what you mentioned about 'a more structured way of addressing token redundancy under high compression'. You can compare CDPruner's method with the fact that it can achieve such high accuracy without spatial coverage. So, what other advantages does your method have?

---

> > > ### Author Response · Authors · 2026-04-03
> > >
> > > Thank you for this important point. We agree that CDPruner represents another effective way to achieve strong performance under high compression, without explicitly enforcing spatial coverage. Our intended claim is therefore narrower: explicit spatial coverage is one effective inductive bias for robust pruning, not the only one. CDPruner optimizes conditional diversity via an instruction-conditioned DPP subset-selection objective, whereas SpecFlow emphasizes structure preservation through graph diffusion and explicit regional allocation.
> > >
> > > Relative to CDPruner, we see three main advantages of SpecFlow. First, it makes the structural prior explicit and spatially interpretable: the retained tokens are region-coherent and the regional budget can be directly visualized and controlled. Second, SpecFlow is a conservative-condensation method rather than pure subset selection: in addition to selecting tokens, it summarizes discarded regions with mean+residual sink tokens, so complementary low-energy context is not dropped entirely. Third, under our added matched video comparison, SpecFlow remains slightly stronger in average performance (63.4 vs 62.5 average Acc.; 3.72 vs 3.68 average Score).
> > >
> > > We therefore do not claim that spatial coverage is necessary for all strong pruning methods; rather, our claim is that it provides a complementary and effective route to robust pruning under high compression. We will revise the paper to state this scope more precisely.

---

### Official Review · Reviewer_tQ3y · 2026-03-12

**Soundness:** 3
**Presentation:** 3
**Significance:** 3
**Originality:** 3
**Overall Recommendation:** 4
**Confidence:** 4

**Summary:**

The paper proposes SpecFlow, a training-free token pruning framework. It computes a stable and smooth score field, then adaptively partitions, prunes, and aggregates the pruned tokens with several sink tokens. The method performs well on both image QA and video QA, while still maintaining a plug-and-play design.

**Compliance With Llm Reviewing Policy:**

Affirmed.

**Final Justification:**

The authors addressed my concerns in the rebuttal. I keep my score as Weak Accept and improve my confidence to 4.

**Key Questions For Authors:**

Please see the weakness section above.

**Limitations:**

yes

**Strengths And Weaknesses:**

Strengths

1. The idea is interesting and relevant. Token pruning is an important problem, and going beyond simple top-K pruning is an interesting direction. Using spectral heat flow for this is also a nice idea.

2. The writing is clear and easy to follow.

3. The results on both image QA and video QA show good performance and efficiency.

Weaknesses

1. The paper introduces two failure modes, spatial fragmentation and loss of contextual evidence. Spatial fragmentation makes sense to me, and Figure 4 does show that the baseline methods are more fragmented. However, for contextual loss, it feels more like an intuition at the moment, without much direct evidence. It would be great to include some examples to better show this failure mode.

2. There seems to be a typo in the caption of Table 1 and 2, where it says video QA.

3. The improvement on video benchmarks seems smaller than on image benchmarks compared with the baselines. Is there any explanation for this? Also, since videos usually have more redundancy, I wonder whether a much larger pruning ratio could be used there.

4. While the method shows nice performance for image QA under different pruning ratios, I am wondering if there is some practical recipe for choosing the pruning ratio in real applications.

5. Since token pruning is mainly for inference efficiency, I am also wondering about the actual inference speed of SpecFlow compared with baseline methods such as HoloV under a similar pruning ratio. Could the introduced modules become a bottleneck for inference speed?

---

> ### Author Rebuttal · Authors · 2026-03-29
>
> We sincerely thank you for the constructive comments on our work. We have thoughtfully considered your comments and respond to each point in turn below.
>
> **1. On evidence for contextual loss.**
> We agree that this point should be demonstrated more explicitly. Our motivation is that attention-only pruning tends to over-concentrate the token budget on a few foreground peaks while discarding surrounding context useful for spatial relations and scene-level reasoning. In contrast, SpecFlow uses energy diffusion, adaptive quadtree partitioning, and crop-wise token allocation to encourage broader spatial coverage and preserve contextual evidence to a greater extent. Our claim is not that every low-energy background token is necessary, but that purely saliency-driven pruning can remove complementary evidence needed for relation-, scene-, and text-sensitive questions. To make this explicit, we have provided additional qualitative examples in the anonymous supplementary link: https://anonymous.4open.science/r/icml7217-C31B/contextual.pdf, including cases where the correct answer depends on background cues or spatial relations.
>
>
>
>
>
>
> **2. On the typo in Tables 1 and 2.**
>
> Thank you catching the caption typo. Tables 1 and 2 are image-understanding benchmarks, not video QA benchmarks, and we will correct both captions.
>
>
>
> **3. On video gains and pruning ratios.**
>
> We believe the smaller relative gain on video mainly comes from the fact that the current version of SpecFlow removes redundancy frame-wise in the spatial domain, rather than explicitly modeling temporal edges across frames, which we already note as a limitation and future direction in the paper. As a result, once strong within-frame redundancy is removed, the remaining errors are increasingly governed by temporal reasoning rather than spatial token selection alone. This likely reduces the relative margin we can obtain on motion-heavy video reasoning compared with image QA, where the main bottleneck is more directly spatial token redundancy.
> Videos contain substantially more redundancy, so more aggressive pruning is feasible. To clarify this, we added a more aggressive video budget sweep. As shown below, SpecFlow remains consistently stronger than FastV and HoloV at both 227 and 114 retained tokens. We are currently exploring temporal-aware propagation and allocation as a natural extension of SpecFlow.
>
>
> | Setting | Method | MSVD-QA Acc. | MSVD-QA Score | MSRVTT-QA Acc. | MSRVTT-QA Score |
> |---|---|---:|---:|---:|---:|
> | Retain 227 Tokens (↓ 88.9%) | FastV | 67.4 | 3.72 | 52.5 | 3.40 |
> | Retain 227 Tokens (↓ 88.9%) | HoloV | 68.9 | 3.87 | 55.4 | 3.46 |
> | Retain 227 Tokens (↓ 88.9%) | **SpecFlow (Ours)** | **69.7** | **3.92** | **55.8** | **3.47** |
> | Retain 114 Tokens (↓ 94.4%) | FastV | 63.2 | 3.62 | 52.1 | 3.39 |
> | Retain 114 Tokens (↓ 94.4%) | HoloV | 65.2 | 3.75 | 53.8 | 3.41 |
> | Retain 114 Tokens (↓ 94.4%) | **SpecFlow (Ours)** | **66.0** | **3.81** | **54.3** | **3.43** |
>
>
> **4. For practical deployment.**
>
> In our view, the pruning ratio should be treated as a deployment knob that reflects the latency–accuracy trade-off of the target application. Based on the current trade-off, our recommendation is: 192 tokens as an accuracy-first default, 128 as the best overall trade-off, and 64 as a latency-first regime with a larger but still graceful degradation. We will add this rule-of-thumb explicitly so that practitioners can choose a pruning ratio based on their target accuracy/latency constraint.
>
>
>
> **5. About inference speed**
>
> We agree that inference speed is critical for pruning research. We report practical efficiency metrics on the A6000 GPU with wall‑clock profiling, including a breakdown of pruning overhead, prefill and end‑to‑end latency. The results show that the added modules introduce only a small overhead and do not become the main runtime bottleneck. Under the same pruning ratio, SpecFlow is slightly slower than HoloV, but remains much faster than the unpruned baseline while achieving better accuracy. We will include these results in the revised version.
>
> | Method | Retained Tokens | Pruning Ratio | Prefill (ms) | Latency (s) | GPU Memory (GB) | Relative Accuracy (%) |
> |---|---:|---:|---:|---:|---:|---:|
> | Vanilla | 576 | 0.0% | 86.6 | 0.397 | 19.2 | 100.0 |
> | FastV | 192 | 66.7% | 42.0 | 0.284 | 16.2 | 90.5 |
> | HoloV | 192 | 66.7% | 22.0 | 0.248 | 15.8 | 98.2 |
> | **SpecFlow (Ours)** | **192** | **66.7%** | **24.5** | **0.259** | **15.9** | **98.7** |
> | FastV | 64 | 88.9% | 29.0 | 0.249 | 15.8 | 76.7 |
> | HoloV | 64 | 88.9% | 14.0 | 0.221 | 14.8 | 94.9 |
> | **SpecFlow (Ours)** | **64** | **88.9%** | **15.5** | **0.229** | **14.8** | **95.6** |
>
>
> | Setting | kNN Graph | Diffusion | Quadtree + TopK + Sink | Total Pruning Overhead | Latency |
> |---|---:|---:|---:|---:|---:|
> | 192 tokens | 1.1 ms | 0.6 ms | 0.9 ms | 2.6 ms | 0.259 s |
> | 64 tokens | 1.1 ms | 0.6 ms | 1.2 ms | 2.9 ms | 0.229 s |

---

> > ### Author Rebuttal · Reviewer_tQ3y · 2026-04-03
> >
> > Thanks to the authors for the rebuttal. The additional results will make the paper stronger. I will keep my initial score.

---

> > > ### Author Response · Authors · 2026-04-04
> > >
> > > Thank you very much for your thoughtful review and positive feedback. We sincerely appreciate your time and effort in evaluating our paper and responses. We are glad that our clarifications helped address your concerns. In the revised version, we will further improve the paper’s clarity accordingly. Thank you again for your valuable comments and suggestions.

---

### Decision · Program_Chairs · 2026-04-30

**Decision:**

Accept (regular)

**Comment:**

This paper proposes SpecFlow, a training-free visual token compression method for vision-language models. It propagates importance scores via spectral heat flow, allocates budgets via adaptive quadtree partitioning, and aggregates pruned information via sink tokens, shifting token pruning from selective retention to information aggregation. Experiments are conducted on multiple image QA and video QA benchmarks, comparing with methods such as FastV, HoloV, and CDPruner, and validating on architectures including LLaVA-1.5, LLaVA-NeXT, Video-LLaVA, and Qwen2.5-VL. At a pruning ratio of 88.9%, LLaVA-1.5 retains 95.6% of its original performance, while inference latency decreases from 0.397s to 0.229s. All four reviewers gave positive final scores (three weak accepts, one accept), and concerns regarding computational overhead and architectural generalization were addressed during rebuttal. Limitations of the method include modest gains over the strongest baseline HoloV in some settings, limited improvement on video tasks, and potential loss of fine-grained information under extreme compression, which the authors discuss in the paper.